

# Impacts of climate and land-surface change on catchment evapotranspiration and runoff from 1951-2020 in Saxony, Germany

Maik Renner[1,2] and Corina Hauffe[1]

[1]Institute of Hydrology and Meteorology, Technische Universität Dresden, Dresden, 01069, Germany
[2]also at, Landesamt für Umwelt, Brandenburg

**Correspondence:** Maik Renner (maik.renner@mailbox.tu-dresden.de), Corina Hauffe corina.hauffe@tu-dresden.de

**Abstract.** This paper addresses the question how catchment scale water and energy balance have responded to climatic and land-surface changes over the last 70 years in the federal state of Saxony, in eastern Germany. Therefore observational data of hydrological and meteorological monitoring sites from 1951-2020 across 71 catchments are examined in a relative water and energy partitioning framework to put the recent drought induced changes in a historical perspective. A comprehensive

visualization method is used to analyze the observed time series. The study focuses on changes in decadal time scale and finds the largest decline in observed runoff in the last decade (2011-20). The observed decline can be explained by the significant increase in aridity, caused by the reduction of annual mean rainfall and an increase in potential evaporation at the same time. In a few mainly forested head water catchments the observed decline in runoff was even stronger than predicted by climate conditions alone. These catchments are still on the recovery from past widespread forest damages in the 1970-80s resulting

in a continuous increase of actual evapotranspiration due to forest regrowth. In contrary runoff stayed almost constant in other catchments despite an increase in aridity. These catchments showed declines in actual evapotranspiration which could be signatures of either contributing groundwater at longer time scales or drought induced vegetation damages.

These results highlight that water budgets in Saxony are in an unstable, non-stationary regime due to significant climatic changes and regional impacts of land-surface changes such as forest health. The recent decreases in the mean annual runoff are

substantial and must be taken into account by the authorities for fresh water management.

## 1 Introduction

Annual mean runoff is an important variable in water management as it determines the available fresh water of a catchment. Runoff is governed by climatic conditions which describe the available water by precipitation and the atmospheric demand for water and by catchment properties such as topography, land-cover, soils and geology (Gentine et al., 2012).

Anthropogenic warming and increasing human pressure on ecosystem services is expected to impact water balance components (Gudmundsson et al., 2021). Recent hydroclimatic trend analysis across Europe (Masseroni et al., 2021) shows continental scale diverging trends for annual mean precipitation with increases in northern Europe and declines in the Mediterranean region. The signal follows the trend pattern of the wet getting wetter and the dry getting drier (Held and Soden, 2006) which was early related to global warming. Although streamflow trends tend to follow the changes in precipitation, there are regions





with diverse trends even opposing the meteorological pattern (Masseroni et al., 2021). Reasons for the diverging pattern of streamflow trends could be manifold: different climatic sensitivities, changes in land-surface conditions (Teuling et al., 2019), seasonal changes with earlier snow melt (Renner and Bernhofer, 2011; Berghuijs et al., 2014), compensating effects, decadal variability influencing trend analyses (Hannaford et al., 2013) but also methodological reasons such as different time windows of available observations.

The eastern part of Germany is such a region where trends in streamflow did not follow trends in climate and where previous research for the federal state of Saxony found large decadal variations, including catchments where major land-cover changes occurred that influenced catchment evapotranspiration and runoff (Renner et al., 2014).

Saxony has a long history of hydrological and meteorological observations which allows hydro-climatic analysis such as the famous work by Schreiber (1904), who established the first type of a Budyko curve also with data from Saxony. Also
Saxony has a wide range of different topographic characteristics from the forested Ore mountains in the south to the flat and arable regions in the northern part. The climate is shaped by the transition of the maritime climate in the west towards more continental climate in the east. Over the last 70 years mean annual air temperature rose significantly by about 1.5 K (Franke and Rühle, 2022) due to global warming by anthropogenic greenhouse gas emissions. While greenhouse gas emissions are still increasing, the emissions of sulphur dioxide were successfully reduced since its peak in 1980s (Maas and Grennfelt, 2016).
Sulfur dioxide has strong environmental impacts. High atmospheric concentrations affected the transmission of visible solar radiation leading to a reduction of surface solar radiation towards the 1990s and an trend reversal since then (Wild et al., 2005). Air pollution was also the main cause of the tree dieback in the mountainous regions of the Ore and Izer mountains (Mazurski, 1986; Šrámek et al., 2008; Maas and Grennfelt, 2016). The forested higher regions were hit more severely since the needle leaf trees comb out fog quite effectively. The fog, which was enriched by sulfuric acid, was intercepted by needle leaf trees
and then infiltrated into soils leading to soil acidification. Over the years the acidic conditions strongly reduced tree growth and caused widespread tree damages (Pitelka and Raynal, 1989; Maas and Grennfelt, 2016). With the reduction of emissions in the 1990s and reforestation activities the forests started to regrow (SMUL, 2006; Šrámek et al., 2008). In recent years forests were also hit by a series of large storms, the 2018-2020 droughts and bark beetle bloom. Here especially monoculture spruce tree forest plantations were hit severely, leading to yet another forest disturbance (Otto et al., 2022).

Renner et al. (2014) analyzed decadal variations in water and energy balance components from 1950 to 2009 and separated the climatic from the land-cover related impacts on evapotranspiration. After 2009 climate change continued and a record breaking drought in Europe occurred (Rakovec et al., 2022; Büntgen et al., 2021). It becomes natural to ask how runoff responded to this drought and to put this change into relation to the earlier changes.

Therefore this manuscript updates the data sources for 71 catchments in the federal state of Saxony and extends the previous
study period to cover 1951 to 2020. The methods section introduces the coupling of the water and energy balances and explains how this coupling can be used to separate climatic from land-cover impacts. Then the dataset is introduced and in the results section the last 70 years are analyzed. The study focuses on the changes from the 2001-10 into the 2011-20 decade which actually showed the largest climatic shift observed in this region. The paper closes with a discussion of the limitations and the impacts of the drought as well as recent land-cover changes are discussed which were partly induced by the drought itself.



## 2 Methods

### 2.1 Catchment water and energy balance

The catchment water balance describes how precipitation $P$ received over the catchment area is partitioned into actual evapotranspiration $E_T$, runoff $R$ at the catchment outlet and changes in water storage $\Delta S_{\mathrm{w}}$:

$$P = E_{\mathrm{T}} + R + \Delta S_{\mathrm{w}} \tag{1}$$

Precipitation, evapotranspiration and water storage in soils and groundwater vary spatially and river discharge integrates runoff processes over the catchment towards the outlet. With the simple water balance equation in (1), we assume that (i) precipitation which is spatially averaged over the catchment area is the only water input into the catchment and that (ii) runoff observed at the outlet comprises all outflow components. The latter two assumptions require that there are no water fluxes, e.g. by groundwater or water management across the catchment area.

Evapotranspiration $E_T$ is also part of the surface energy balance, here normalized by the latent heat of vaporisation $L = 2.5 MJkg^{-1}K^{-1}$:

$$R_{\mathrm{n}}/L = E_{\mathrm{T}} + H/L + \Delta S_{\mathrm{e}}, \tag{2}$$

with net radiation $R_{\mathrm{n}}$, sensible heat $H$ and an energy storage change term $\Delta S_{\mathrm{e}}$. Since available energy usually expressed as $R_{\mathrm{n}}/L$ is not well observed, we assume that it can be described by potential or reference evapotranspiration denoted as $E_0$ (Choudhury, 1999; Arora, 2002).

$E_T$ couples the water and energy balances. As first consequence of the coupling we can state that $E_T$ is limited by both precipitation and available energy at the same time (Budyko, 1948). Thereby $P$ and $E_0$ become the two main predictors of the water and energy balance, which is well documented by the Budyko curve (Budyko, 1948). It describes how the relative water balance, here $E_T/P$, can be well predicted by the aridity index $E_0/P$. The semi-empirical functions of Schreiber (1904); Ol'Dekop (1911); Budyko (1948); Pike (1964) and others show good agreement with catchment water budget data from humid to arid climates. The relative water balances of most catchments are close to the physical limits. However, variation in regional catchment properties may affect the partitioning. For example steep orography or seasonality of precipitation may decrease $E_T$, while density and type of vegetation, plant accessible soil water tend to increase $E_T$ under the same meteorological forcing. These effects were summarized in a catchment parameter from which parametric Budyko curves were developed (Turc, 1961; Mezentsev, 1955; Fu, 1981).

The well-known way to visualize the water and energy coupling is the Budyko plot. The aridity index $E_0/P$ is plotted on the x-axis and the relative water partitioning $E_T/P$ on the y-axis. The later ratio should range between 0 and 1. Low values indicate that most of the rainfall is converted into runoff and the upper limit of 1 would indicate that all of the received precipitation evaporates.



## 2.2 Decomposition of climatic and land-surface impacts

The second consequence of the complementarity of water and energy balances is that changes in the aridity of a catchment will not only change the relative water balance $q = E_T/P$ but also the relative energy balance $f = E_T/E_0$, only in the opposite direction. This is illustrated in Fig.1 which plots the relative water balance on the x-axis and the relative energy partitioning on the y-axis. Any point in the range between 0 and 1 is a reasonable state of the relative water and energy partitioning. The diagonal represents the condition when the aridity index $E_0/P = 1$, with humid conditions above and arid conditions below the diagonal line.

The complementarity becomes apparent under a change in aridity. For example more precipitation will decrease $E_T/P$ but increase $E_T/E_0$, if the catchment follows a Budyko type of curve. In contrast, if there is solely a change in the catchment attributes without variation in the aridity index there will be a parallel shift to the line of constant aridity through the origin. An observed change in $E_T$ (from point $q_0$, $f_0$ to point $q_1$, $f_1$) can be decomposed into a land surface change part (from point $q_b$, $f_b$ to $q_0$, $f_0$, red arrow in Figure 1) and a climate change part (from point $q_1$, $f_1$ to $q_b$, $f_b$, blue arrow). Renner et al. (2014) assumed that climatic changes lead to a shift in $q$ and $f$ perpendicular to the line of constant aridity. With this assumption the amount of evapotranspiration $E_{T,b}$ without any climatic changes can be computed from data before (index 0) and after the change (index 1):

$$E_{T,b} = P_0(f_0 f_1 q_0 + q_0^2 q_1)/(f_0^2 + q_0^2). \tag{3}$$

Then the climate-induced change in evapotranspiration can be calculated by $\Delta E_{T,C} = E_{T,1} - E_{T,b}$, as well as the remaining part denoted as the the land-surface induced change $\Delta E_{T,L} = E_{T,b} - E_{T,0}$.

By applying the water budget equation to the changes, i.e. $\Delta P = \Delta E_T + \Delta R$, we can also compute the amount of runoff change attributed to climatic changes:

$$\Delta R_C = R_1 - P_0 + E_{T,b}, \tag{4}$$

where, $R_1$ is the mean annual runoff in the second period and $P_0$ is the mean annual precipitation of the first period. The land-surface induced changes in runoff are derived by

$$\Delta R_L = P_0 - E_{T,b} - R_0. \tag{5}$$

where $R_0$ is the mean annual runoff of the first period. Note, that changes in runoff are mainly driven by changes in precipitation (Dooge, 1992) and secondly by changes in evaporative demand and land-surface conditions.

Other approaches to decompose climate and land-surface impacts on runoff were proposed by Wang and Hejazi (2011); Jaramillo et al. (2013); Jaramillo and Destouni (2015). These methods use Budyko functions to estimate the climatic change part. However, the uncertainties in quantifying catchment precipitation, runoff and energy demand are more relevant than the methodological differences.





## 2.3 Comprehensive visualization method

The decadal changes in catchment evapotranspiration are related to precipitation and potential evapotranspiration and plotted together in the joint water-energy balance diagram. This allows a visual interpretation of separate climatic and land-surface impacts on catchment evapotranspiration as illustrated in Fig. 1. Therefore decadal averages of the relative water ($E_T/P$) and energy ($E_T/E_0$) partitioning for each of the 71 catchments are calculated. Subsequently the decadal averages of both ratios for a given decade and each catchment are plotted in the joint water-energy balance diagram. The position of a catchment within the plot is depicted as pie-chart. It holds information about the land-use types (see chapter 3.3) of each catchment.

The change of the relative water ($E_T/P$) and energy ($E_T/E_0$) partitioning towards the following decade is integrated as an arrow. The direction of the arrow reveals, if only climatic or land-surface changes occurred or if both types of changes were present. The arrow length refers to the magnitude of the change. In order to assess if a change is statistically significant a two sample Hotellings T2 test (Todorov and Filzmoser, 2009) is performed by sampling the annual data of each decade. Black bold arrows highlight if the magnitude of a change in the decadal mean between two decades is larger than the year-to-year variability using a significance level of $\alpha = 0.1$. Gray thin arrows visualize the direction of an insignificant change.

Furthermore, a map is added with the outline of the analyzed Saxonian catchments to illustrate where the changes occurred. The arrows are also included in the map and start at the river gauge location. This is combined with spatial information of forest damage data (see chapter 3.3).

If several of these comprehensive visualization diagrams are plotted as sequence over a number of decades, it provides information about temporal as well as spatial changes in the catchment water balance. It allows the identification of general behaviour within a large catchment sample and may also reveal differences for certain catchment groups. At the same time the analysis of probable causes is possible.

## 3 Data

In this section details about the used datasets for the hydro-climatological analysis are provided. Note that this study updates the analysis presented by Renner et al. (2014) by adding data from 2010-2020. Thereby the same catchments and methods are used as in Renner et al. (2014). However, the hydrological and meteorological input data were updated containing data corrections and additional meteorological data due to digitization efforts.

## 3.1 Hydrological data

The study covers data of 71 river gauging stations with long term, high quality data records. The sites where chosen according to continuity, homogeneity, small direct impacts by water management due to dam operation or large mining activities such as in the Lausitz region in the East of Saxony. The sampling of the catchments thus slightly over-represents the higher regions of Saxony. It is also important to note that the sample includes nested catchments which share parts of the catchment area and are thus not independent. All catchment attribute data such as river names, catchment polygons etc were used from Renner





et al. (2014). The map in Fig. 2 provides an overview. All time series data were updated. Key data provider is the Saxon State Department for Environment, Agriculture and Geology (Landesamt für Umwelt, Landwirtschaft und Geologie, LfULG), which provided daily runoff time series data via the webportal ida.sachsen.de and by e-Mail. The river gauge Greiz/Weiße Elster is operated by the Thuringian State Department of Environment, Mining and Nature Conservation (TLUBN) and time-series data

were obtained directly from the authority. All data were checked graphically for correctness. Data screening for homogeneity and dominant influences like water management was performed in Renner et al. (2014) and such catchments were removed from the dataset.

At first daily discharge data were aggregated to monthly averages with at least 20 non-missing values per month. Then annual averages were computed using the hydrological year from November to October. Finally decadal averages and statistics

were derived when at least 6 years of complete data per decade were available. Runoff yield ($mm\ yr^{-1}$) was calculated from average discharge records ($m^3\ s^{-1}$) by multiplying with the number of seconds per year and dividing by the catchment area.

The study period covers 1951-2020 and for computing the long-term average runoff use all available annual data per site were used. Note that not all runoff time series cover the whole period. In a few catchments river gauge relocation occurred, leading to a change of the station-id. At the following sites the time series of the prior gauge and the current station were

merged for this study: Buschmühle/Kirnitzsch, Piskowitz 2/Ketzerbach and Annaberg/Sehma. Table 1 provides an overview of the river gauges and the time series availability.

### 3.2    Meteorological data

Precipitation and potential evaporation were derived from meteorological station data, interpolated on a spatial grid, from which catchment averages were extracted. Precipitation is well observed in the study area with 845 sites used in the regionalisation and

366 sites with more than 40 years of data, shown as circles in Fig. 3a. The size of the circles reflects the available observation years. Key operators are the German weather service, the Landestalsperrenverwaltung Sachsen (river dam authority of the Federal State of Saxony) as well as the Czech and Polish Hydrometeorological Institutes. All meteorological station data were acquired at daily resolution from the REKIS (rekis.org) data service, which is a platform for regional meteorological data service covering the Federal States of Saxony, Thuringia and Saxony-Anhalt.

The original daily precipitation data was added up to annual totals using the hydrological year from November to October. The annual station data were then interpolated on a 1.5 km grid using an automatic universal Kriging procedure provided as R package automap (Hiemstra et al., 2009). Thereby station altitude is used as the dependent variable. The procedure was validated by Renner and Bernhofer (2011) and Renner et al. (2014).

To model potential evaporation ($E_0$) we use the FAO-56 method for grass reference evapotranspiration (Allen et al., 1994).

This is a simplification of the Penman-Monteith equation and Allen et al. (1994) provide methods to substitute net radiation data and ground heat flux which are not commonly available. Here, the daily station data was first aggregated to monthly averages to compute $E_0$ from temperature (daily mean, minimum, maximum), sun shine duration, relative humidity and wind speed data. The locations of the 115 climate stations (40 sites have more than 40 years of data) are shown as circles in Fig. 3b. Similar to





precipitation spatial interpolation was performed on available annual (hydrological year) station data and interpolated onto a
1.5 km grid using universal Kriging with altitude as dependent variable.

### 3.3  Land-use and forest damage data

For the assessment of the dominant land-use types the remote-sensing based Corine Land Cover data set (© European Union,
Copernicus Land Monitoring Service, European Environment Agency (EEA)) is used with snapshots from 1990, 2000, 2006,
2012. From the 2000 snapshot the land-use types of forested and near natural vegetation classes as "forest", all agricultural
and grass lands are merged as class "agricultural" and the remaining as "others", mainly built-up areas. To derive the area of
damaged forest within a catchment, the Corine land cover class Transitional Scrub-Forest (324) is used. This land cover class
includes areas of damaged forests (Bossard et al., 2000). For the pre-satellite era maps of canopy damage data available for the
years 1960, 70, 80 and 1990 were used. Canopy damage was assessed by needle and leaf losses in the canopy (SMUL, 2006)
and follows a measurement protocol defined in the former German Democratic Republic (Forstprojektierung, 1970).

## 4  Results

### 4.1  Long-term hydro-climatology of Saxony

In this section the long-term average (1951-2020) hydro-climatology of Saxony is illustrated based on the established dataset,
see also Table 1.

Annual average precipitation is $830\,\text{mm yr}^{-1}$ across the 71 catchments and ranges between 607 and $1151\,\text{mm yr}^{-1}$. For the
whole of Saxony an average of $712\,\text{mm yr}^{-1}$ is estimated. The difference is due to the sampling of the catchments, which
over-represent higher regions of Saxony. Precipitation shows a distinct north to south increasing gradient which is linked to
altitude. Another, although weaker gradient is induced by the transition of maritime to continental climate which results in
decreased precipitation from west to east (Fig. 3a).

The FAO grass reference potential evapotranspiration $E_0$ across the study catchments ranges between 572 and $738\,\text{mm yr}^{-1}$
and is on average $669\,\text{mm yr}^{-1}$. $E_0$ is negatively correlated with elevation (see Fig. 3b).

The spatial pattern of runoff is dominated by topography and precipitation with most catchments receiving a large amount
of its water from the headwater catchments in the Ore Mountains, the Lausitzer Bergland, and Izer mountains, see Fig. 3c.
The lowest annual runoff values are found in the lowland catchments in the North (minimum $59\,\text{mm yr}^{-1}$) and highest in the
headwater catchments in the South (maximum $799\,\text{mm yr}^{-1}$). On average runoff is $372\,\text{mm yr}^{-1}$ across the 71 catchments.

The difference of $P - R$ is used to estimate actual catchment evapotranspiration $E_T$ (Fig. 3d). $E_T$ is on average of all
catchments $461\,\text{mm yr}^{-1}$ and $572\,\text{mm yr}^{-1}$ on maximum. The lowest values of annual $E_T$ are found in the high headwater
basins (minimum $219\,\text{mm yr}^{-1}$). In general $E_T$ has a smaller spatial variation than runoff. All long-term average data is
summarized in detail in Table 1.





The average water and energy partitioning of the catchments in Saxony is illustrated in Fig. 4a) and Fig. 4b), which is
the Budyko plot. Each catchment is depicted as pie chart, representing the relative portion of forested and agricultural area.
Furthermore contour lines are included in Fig. 4a), derived from the mean catchment altitude.

The relative water partitioning $E_T/P$ is placed on the x-axis in Fig. 4a) as well as on the y-axis in Fig. 4b). The long-term
averages of $E_T/P$ vary between 0.22 and 0.91 and are on average 0.57.

On the y-axis of Fig. 4a) the relative energy partitioning $E_T/E_0$ is plotted, also ranging between 0 and 1. This ratio varies
between 0.37 and 0.83, with most catchments clustering around the average value of $E_T/E_0 = 0.69$. Plotting both ratios
together combines the two key meteorological forcing variables $P$ and $E_0$. Three general examples for the aridity index
$\phi = E_0/P$ are included in Fig. 4a) as lines through the origin. When precipitation equals potential evaporation $P = E_0$ this
would fall on the diagonal. More humid conditions would be above the diagonal of $\phi = 1$, more arid conditions fall below this
diagonal. All together catchments at higher altitudes (see altitude contour lines in Fig. 4a) are more humid with $E_T/P < 1$
and have a larger portion of forested area. Catchments with $E_T/P > 1$ are generally found at lower altitudes and dominated
by agriculture land use.

The aridity index $\phi$ is also visualized on the x-axis in the Budyko plot (see Fig. 4b) and ranges between 0.50 at the highest
catchment and 1.21 at the lowest catchment. On average we find a value of $E_0/P = 0.82$. The Budyko function (Budyko,
1948) is plotted as reference allowing the prediction of catchment $E_T$ from the aridity index. The distribution of the data
generally agrees with the Budyko function. However, there is an overestimation of $E_T/P$ at the more humid catchments and
an underestimation for the drier catchments.

## 4.2 Annual and decadal dynamics in joint water and energy balance

### 4.2.1 Variability of meteorological forcing

The variability of the meteorological forcings $P$ and $E_0$ will be analyzed using the annual mean values as well as the decadel
means (see figure 5). Generally there is a larger variability of precipitation $P$ both across catchments and from year to year
than $E_0$. The decadal mean precipitation over all basins was always higher than of $E_0$. However, the last decade (2011–20)
shows a remarkable increase in $E_0$, being the highest decadal average so far (710 mm/yr) and getting closer to $P$ (814 mm/yr).
At annual time scale about 17% of the years had a negative climatic water balance, where catchment annual average $E_0$ was
larger than $P$: 1951, 1953, 1959, 1963, 1964, 1976, 1982, 1990, 1991, 2003, 2015, and 2018, depicted as circles in figure 5.
The occurrence of these negative values does not have a clear trend, but indicates that there have already been dry conditions
in the 1950s and 1960s.

### 4.2.2 Variability of catchment evapotranspiration

The water budget residual $P - R$ is analyzed as proxy for basin scale $E_T$, which may include the unknown influence of past
water storage changes $\Delta S_w$. To reduce the effect of storage changes decadal averages of $E_T$ are derived, presented in Fig. 6. In
the figure catchments are grouped by the percentage of areal forest damage extent. Therefore the transitional scrub-forest class





in the Corine data set of the year 1990 is used, following Renner et al. (2014). The snapshot of 1990 was used since reliable Corine data was available during the period of maximal forest damage extent. Based on this data, 38 % of the catchments have no forest damage ($< 0.01$ %), 21 % have minor affected areas (0.01–2 %), 31 % have considerable damaged areas (2–20 %), and 10 % of the catchments have damaged areas larger than 20 %. The respective catchments are shown in the map in Fig. 7.

250      The boxplots in Fig. 6 depict variation of the decadal average of $E_T$ of all catchments in each group. During the first decade 1951–60 there is no large difference between these four groups. Between 1961–80 the heavily damaged group experienced a large decline in $E_T$, while the two groups with low damage effects show little variation. The catchments with moderately damaged forest are in between and show a minor decrease in $E_T$ during the decades from 1971–90. From 1991–2010 all catchments have a pronounced increase in $E_T$. And while the positive trend in $E_T$ continues for the group with considerable 255    forest damages, the low damaged catchments show a small but consistent decline during the last decade.

### 4.3    Decomposition of climate and land-surface effects

The changes in runoff are decomposed in a climate and a land-surface change component which together sum up to the observed change. The decomposition method by Renner et al. (2014) introduced in the methods section quantifies the contributions of climatic changes in $P$ and $E_0$ as well as land-surface changes in $E_T$ in retrospective. Figure 8 presents the results of the 260    decomposition for catchment runoff. The gray box-plots depict the changes in the observed runoff from one decade to the next for all the catchments. There is not much variation of the median of the changes from 1951 to 2010 and no clear tendency for the direction of change. The observed changes in runoff varied between -50 and 50 mm/yr across the Saxonian catchments. In the last decade observed runoff decreased the most across the majority of catchments with a median decline of -70 mm/yr, which is 8% of mean annual precipitation.

265      The decomposition quantifies the contributions of climatic changes in $P$ and $E_0$ on runoff, visualized as blue box-plots in Fig. 8. Thereby the climatic driven runoff changes were relatively small in the first five transitions with median changes between -13 mm/yr towards the 80s and +23 mm/yr towards the 2000s. The climate driven changes became the dominant cause in the last transition with a median change of -65 mm/yr and thus explaining the observed decline in runoff in most catchments in Saxony. However note, that the transitions from the 80s to the 90s and from the 90s to the 2000s showed two 270    consecutive increases in the climatic runoff change component due to increases in precipitation.

     The red box-plots in Fig. 8 represent the land-surface change component. The outliers show that in a few catchments it dominated the change in runoff during the past decades. There have been two consecutive, but small increases of 14 mm/yr in runoff due to land-surface conditions towards the 70s and 80s. They were followed by a strong and more widespread decline in the 1990s with a median decrease of -38 mm/yr in the land-surface component.

275    ### 4.4    Decadal dynamics of water-energy partitioning from 1951 to 2000

The decadal changes within the 70 year period are depicted by six panels in Fig. 9. In each panel the changes in the water and energy partitioning ratios from one decade to the next is visualized by the direction of the arrows as well as the length. The





displayed pie-charts always refer to the same land-use types derived from the Corine dataset of the year 2000. The number of catchments taken into account varies due to the different availability of data.

Panel a) shows a set of 28 catchments in the decade 1951-60 with transition into the next decade. No significant changes in $E_T/E_0$ and $E_T/P$ are apparent. In Fig. 9b) the transition from the 60s to the 70s is visualized. The number of available catchments increases to 45. The changes within the catchments are either pointing towards the origin of the diagram or in the opposite direction, but are not significant for most of the catchments. Only two neighboring head water catchments in the upper Ore mountains (Rothenthal/Natschung and Zöblitz/Schwarze Pockau) have significant decreases in both ratios indicating an impact of land-surface changes on $E_T$. Also other neighboring catchments show decreases which hints at a similar cause. These declines in $E_T$ are mostly in the more humid head water catchments and are part of the group with dominant forest damage, as presented in Fig. 6.

The extent and severity of forest damages increased in the 1980s, see inlay maps of forest damage. The transition from the 1970s to the 80s was analyzed for 69 catchments and can be seen in Fig. 9c). The development of the two ratios appears to be dominated by this ongoing spread of forest damages, with many of the arrows showing directions of land-surface changes. For two catchments in the upper Ore mountains (Rehefeld/Wilde Weisseritz and Wolfsgrund/Chemnitzbach) the declines in $E_T/E_0$ and $E_T/P$ are significant, while four catchments experience a significant increase in the two ratios.

In contrast to the prior transitions the changes are almost consistent for all of the catchments in the transition from the 80s to the 90s depicted in Fig. 9d). It reveals the most drastic and coherent shifts observed in the whole period, with dominant and significant increases in both ratios in 11 out of 69 catchments. For some of the catchments this even means a reversed direction of changes for $E_T/E_0$ and $E_T/P$ compared to earlier periods. In 1990 forest damage reached its largest extent in Saxony. In addition increases in precipitation and potential evaporation led to a further increase in catchment evapotranspiration, see Fig. 5. The following decadal change from 1991–2000 into 2001–2010 is visualized in Fig. 9e). Now there are two groups with a different development in the water and energy partitioning. While the mountainous catchments, which were highly effected by the vast forest damages, still have a continuing increase in $E_T/E_0$ and $E_T/P$, the lower-lying regions generally show less changes with no clear direction. Fig. 9e) also reveals five mountainous catchments with significant recovery of both ratios. In contrary the lowland catchment Seerhausen/Jahna has a significant decrease in the aridity index due to higher precipitation.

In general the land-surface attributed impacts were dominating the development of the two ratios $E_T/E_0$ and $E_T/P$ for the time period between 1951 and 2010. This analysis has been part of the previous work of Renner et al. (2014). The picture changes, when it comes to the transition from 2001–10 to the last fully observed decade of 2011–20, which is presented in the next section.

## 4.5 Decadal dynamics of water-energy partitioning from the 2000s to the 2010s

The last decadal transition is dominated by the meteorological conditions of 2011-20, which have been much drier than the two decades before. This results in the highest number of significant changes (16 out of 64, 25%) in catchment water and energy partitioning, presented in Fig. 9f). The direction of change is now mainly orthogonal to the prior transitions, meaning that climate attributed impacts effecting the ratios of $E_T/E_0$ and $E_T/P$ the most.





Focusing at the significant changes three different types can be observed: (i) five catchments in the Ore mountains show
continuing recovery effects with significant increases in both ratios. These catchments have had large areal extents of forest
damages in which the recovery led to catchment evapotranspiration rates now being even higher than during the 1951–60

decade, as can be seen in Fig. 6. Type (ii) are catchments, especially in the Mulde river basin, which experienced quite sig-
nificant decreases in annual mean precipitation. Nine of the catchments show significant movements in the climatic change
direction. These catchments follow the Budyko curve and retain high evapotranspiration leading to increases in the relative
water balance. Type (iii) are the low-land catchments in northern and eastern Saxony recognizable by the downward pointing
arrows in Fig. 9f). They represent a decrease in the relative energy partitioning without a change in the relative water balance.

Only one catchment (Zescha/Hoyerswerdaer Schwarzwasser) experienced a significant change. Here the increase in potential
evaporation did not lead to an increase catchment evapotranspiration. This highlights that both, climate changes (i.e. higher
aridity) and land-surface changes, co-occurred in this transition period. A similar behavior was also found by Pluntke et al.
(2023) for a small, forested research catchment in the south of Dresden, Saxony.

Two more figures are included to illustrate the noticeable changing conditions from 2001–2010 to 2011–20 in Saxony. Fig. 10

visualizes the decadal changes as bar plot for all analyzed water balance components and all 71 catchments. The reduction in
precipitation (blue bars in Fig. 10) on the one hand and the uptake in the potential evaporation (red bars in Fig. 10) on the other
hand are apparent. The widespread and large decline in annual precipitation with a median decrease of -59 mm/yr. It was rather
pronounced in the Ore mountains reaching up to -143 mm/yr and the Izer mountains, which feed the upper Lausitzer Neiße
river, see Fig. 11a). The maps in Fig. 11a) combine precipitation changes (filled polygons of the catchment area) with changes

in observed runoff (filled triangles at the river gauge locations). The same color scale is used to highlight, how much the
regional precipitation decrease affected runoff. This combination shows that catchment runoff broadly followed precipitation.
In 56 out of 64 catchments the observed runoff decreased with a median decline of -73 mm/yr.

The map in Fig. 11a) also highlights notable exceptions: (i) there are a few catchments in the Upper Ore mountains with
larger decreases in runoff than in precipitation, and (ii) four catchments located in the East of Saxony having notable increases

in runoff despite increasing aridity.

In addition to less precipitation amounts there was a substantial increase in FAO reference potential evaporation across all
regions with a median increase of 36 mm/yr, see red bars in Fig. 10. The map in Fig. 11b) presents a rather uniform spatial
distribution of the change in potential evapotranspiration with slightly larger increases in lower parts of Saxony. In this map
the changes in actual catchment evapotranspiration $E_T$ (derived by $P - R$) are added with the same color scale. The response

of actual evapotranspiration is rather mixed. 27 out of 64 catchments depict a reduced $E_T$ with -10 to -67 mm/yr, see also the
orange bars in Fig. 10. These catchments are found in the lower regions of Saxony. Especially the smaller tributaries to the Elbe
River and catchments in the Lausitz show these reductions (triangles in Fig. 11b). Furthermore, 16 catchments experienced
increases in $E_T$ larger than 10 mm/yr. In five of these catchments in the upper mountain regions the increase even exceeded 50
mm/yr. The remaining 21 catchments have only smaller changes in $E_T$.

The decomposition method was applied to quantify the climatic and the land-surface impact on runoff changes. The climatic
impact of the observed changes in precipitation and potential evaporation on runoff is visualized in the map in Fig. 11c). The



catchments in the Mulde river basin show the largest decreases in the middle of Saxony and there are also larger effects seen in the upper Lausitzer Neiße river basin. Note that these catchments have a more humid climate and higher runoff ratios, which generally leads to a higher sensitivity of absolute runoff amount to changes in precipitation (Dooge, 1992; Renner et al., 2012).

In the decade of 2011-2020 the increasing dryness has had the strongest climate impact on runoff in Saxony in the whole period analyzed here (see Fig. 8).

Fig. 11d) shows the spatial distribution of the land-surface impacts on runoff. Many catchments reveal only minor effects of land-surface changes. However, there are interesting exceptions. The group of catchments with the large increases in $E_T$ and corresponding pronounced declines in runoff is the one with the most land-surface impacts. The catchments are all lo-

cated in forested regions, where the recovery of past forest damages is still ongoing (Ore Mountains, the Lausitzer Bergland) as presented in Fig. 11d). There are also catchments with increasing runoff attributed to land-surface impacts as shown in Fig. 11d). These are predominantly low-land catchments with higher aridity indices and no decrease in runoff as expected from the climatic conditions, resulting in downward arrows in Fig. 9f). The discussion of the potential causes is left for the next section.

## 5 Discussion

In this study we identified and estimated impacts of climate and land-surface conditions on catchment water balance in the Federal State of Saxony. In the following potential limitations of the study are explained. Afterwards the role of climatic impacts as well as changes in the land-surface on catchment water balance is discussed.

### 5.1 Limitations

### 5.1.1 Catchment selection and influences by water management

The selection of catchments aimed to capture a large part of Saxony, while excluding catchments with direct hydrological impacts such as reservoirs with storage volumes large enough to influence the annual water balance. Also catchments which are impacted by large water extractions and inputs have been excluded. These exclusions were done using meta data as well as homogeneity tests as explained in Renner et al. (2014). Nevertheless, there may still be artifacts in single catchments due to

these issues. For example the catchments Rehefeld/Wilde Weisseritz, Baerenfels/Pöbelbach and Geising/Rotes Wasser border to the catchment of the drinking water reservoir Altenberg which was established in the 1980s. The operation of trenches could thus directly influence the water budget of these catchments. Since the reservoir is not directly part of these catchments we left them in the dataset, but excluded river gauges whose catchments is dominated by reservoirs, such as the Rote Weisseritz river. For a full treatment of catchments, which are affected by water management there is the need to collect and share data on

reservoir operation, water abstractions and transitions.





### 5.1.2 Catchment integration and groundwater transfers

Closing the water balance is a key assumption to estimate catchment evapotranspiration. Water fluxes across catchment boundaries, for example by groundwater flows contributing to downstream parts of the river basin (Fan, 2019), were not considered in this study. Both natural and anthropogenic cross boundary fluxes are often difficult to quantify and thus lead to biases in

$E_T = P - R$ and especially absolute values must be taken with care. However, a reasonable assumption is that unmeasured water fluxes across catchment boundaries may not change over time. Thus the decadal change signals in runoff and catchment evapotranspiration are likely unaffected by this type of uncertainty. Nevertheless we recommend to address this issue in further studies which explicitly include groundwater fluxes and water transfers across surface catchment boundaries.

### 5.1.3 Runoff estimation

Runoff was computed from daily discharge time series data obtained from the monitoring authorities. Discharge is usually estimated from continuous water level records and a water level – discharge relationship. It is established by point measurements usually done every one or two months. These relationships can change over time due to changes in the profile, such as weed growth and weed control but also by flood deposits or ice conditions. Hence runoff estimates can be quite uncertain under low flow conditions as well as under flood conditions when the river leaves its profile (Coxon et al., 2015). By averaging

the impact of these uncertainties they can be reduced but not completely removed. Again the large sample of catchments analyzed here shows that there are consistent changes across catchments. It highlights that this uncertainty is not a major issue in interpretation of the results.

### 5.1.4 Temporal resolution

The analysis is based on annual averages and thus neglects any subseasonal changes such as hydro-meteorological shifts in

spring season (Bernhofer et al., 2008; Renner and Bernhofer, 2011; Berghuijs et al., 2014; Ionita et al., 2020). These seasonal shifts are out of scope of this study and should be investigated in further studies. Further decadal averages are used for the decomposition of climate and land-surface impacts to reduce the potential impact of interannual storage changes, which can be large in groundwater dominated catchments. Changes in storage, if present at the interdecadal time-scale, would certainly influence the climate and land-surface decomposition, since a storage change would be mistaken for a change in evapotranspi-

ration, see Eq. 1. This potential misinterpretation must be kept in mind when applying the decomposition framework. However, the results of this study, providing spatially and temporally consistent trajectories in the relative water and energy balances, indicate that storage influences are a minor issue.

## 5.2 Climate change becomes a significant driver of water balance

The drought from 2018–20 in Europe was exceptional in terms of spatial extent, duration and intensity (Rakovec et al., 2022;

Büntgen et al., 2021). Compared to previous drought events, it was exceptionally warm (Rakovec et al., 2022) and was probably caused by anthropogenic warming (Büntgen et al., 2021). The drought assessment of Rakovec et al. (2022) used a modeled





soil moisture index based on meteorological data reconstructions from 1766 to 2020 for Europe. According to Rakovec et al. (2022) northern Saxony and the Lausitz region exhibit a soil moisture drought of 24 months. Low flow statistics showed long duration of hydrological drought conditions which were also exceptionally low in Saxony. The Saxon State Department of
Environment, Agriculture and Geology reported that during summertime of 2018–19 and 2020 40 % to 70 % of 146 runoff gauges in Saxony showed values below the mean annual low flow (Franke and Rühle, 2022). Due to the long duration also mean annual flows of 2018-20 were lower than the long term mean (Franke and Rühle, 2022).

The widespread runoff decline in Saxony represents the largest and most coherent decadal shift in runoff and the decomposition methods highlight that this reduction is caused by climatic changes - that is the reduction of rainfall and the increase
in atmospheric demand for water, see Fig. 8. For the majority of catchments the reduction of runoff can thus be predicted by knowing the changes in annual mean precipitation and potential evaporation. However, there were notable exceptions in certain regions which are discussed next.

### 5.3 Land-surface impacts on water and energy partitioning

Any change in runoff which cannot be predicted with the climatic signal is attributed to land-surface changes. In the last decade
with the shift towards more arid conditions we observed two different reactions in catchment runoff which were attributed to land-surface impacts. While some catchments showed stable or even increasing runoff, others experienced very strong declines in runoff which could not be explained by the dryness, see section 4.5.

#### 5.3.1 Potential causes of constant or increasing runoff

At first sight it seems contradictory that runoff remained stable or increased even though precipitation decreased and potential
evaporation increased. A potential hypothesis explaining this behavior may be the reduced transpiration and evaporation from interception due to reduced growth or plant damage. This hypothesis is supported by forest inventory data from Saxony (Otto et al., 2022) showing that by far the largest amount of damaged wood due to bark beetle infestations occurred in 2018-20 within more than 70 years of observations. The combination of warm and dry conditions reduced the fitness of needle leaf trees, which than led to an outbreak of bark beetles in 2018 continuing to 2020 throughout Saxony. Thereby the forests in eastern Saxony
were most affected, which is inline with regions were we observed stable or increasing runoff conditions and thus decreases in catchment actual evapotranspiration, see Fig. 11.

Apart from forests, there is evidence of reduced crop yields during the drought year 2018 (Statistisches Landesamt Freistaat Sachsen, 2022). This year showed the highest aridity index in the analysed period. The reduction in yields corresponds to the depletion of soil moisture during the long and severe drought, especially in northern and eastern Saxony (Rakovec et al.,
2022). Water limitation causes reduced plant growth as well as reduced transpiration via stomata closure (Teuling et al., 2010). However, more research is required to quantify the contribution of reduced transpiration of short vegetation such as grass and crops to reduced catchment evapotranspiration.

In either case a future recovery of vegetation, especially of forests, will likely increase $E_T$ and thus lead to less runoff from such catchments if precipitation will remain low.





### 5.3.2 Hydrological response to forest recovery under dry conditions


Large scale forest dieback due to effects of air pollution occurred in Central Europe with the industrial use of lignite coal burning with significant regional impacts starting in the 1960s (Mazurski, 1986; Maas and Grennfelt, 2016). Renner et al. (2014) documented on the one hand that these forest damages led to strong reductions in catchment actual evapotranspiration lasting over two to three decades and on the other hand that an increase in $E_T$ correlates with recovery of forests. This link of

forest damage and substantial declines in catchment evapotranspiration is updated in this study and clearly revealed by decadal averages of $E_T$ (see Fig. 6). The effect becomes apparent when the catchments are grouped by the percentage of forest damage.

Since 1990 the recovery of these areas led to increases in catchment evapotranspiration and reduced the gap between disturbed and undisturbed catchments in Saxony. Nevertheless, the analysis of climatic and land-surface attributed changes on runoff, visualized in Fig. 9f) and Fig. 11d), point out that in a few mountainous catchments recovery of $E_T$ is still ongoing.

The largest increases were seen in the Eastern Ore Mountains at the neighboring catchments Baerenfels/Pöbelbach and Rehefeld/Wilde Weißeritz, followed by the Middle Ore Mountains (Rauschenbach/Rauschenfluß, Deutschgeorgenthal/Rauschenbach) and a head water catchment in the Western Ore mountains (Sachsengrund/Große Pyra). At these catchments the strong increases in $E_T$ correlate to exceptional decreases in annual mean runoff. Hereby it is noticeable that the change in runoff is larger than the observed decrease in precipitation. Furthermore, the increase in $E_T$ is higher than in $E_0$ (see Fig. 10). This

strongly hints at the recovery of transpiration and evaporation from interception due to vegetation regrowth. The long term time series of catchment evapotranspiration presented in Fig. 6 shows that the recovery in terms of $E_T$ in these catchments is almost completed. On the one hand the magnitude of $E_T$ is now at the same level or even higher than prior to the large scale forest damages. On the other hand $E_T$ is now in a similar range compared to catchments without severe forest damages.

### 5.4 Comparison of decomposition methods

Last but not least the decomposition method itself is discussed. For validation of this method the results are compared with the algorithm presented by Wang and Hejazi (2011). They employ a parametric Budyko function, here the form of Mezentsev (1955), to estimate the climatic and the land-surface change part. First they estimate the catchment parameter $n$ using the data of the first period. Then the parametric Budyko function is used to derive $E_T$ based on precipitation and potential evaporation of the second period. The difference to the observed change in $E_T$ is the land-surface induced change on $E_T$. The input

data to both methods is the same. One key difference is that the method of this study uses a linear approximation whereas Wang and Hejazi (2011) employ a non-linear Budyko function, which addresses problems in this non-linear space for example when approaching the physical water and energy limits. A second, probably more relevant difference is that Wang and Hejazi (2011) first computes the climatic part while this study derives the land-surface part first, see Fig. 1. Fig.12 illustrates the differences between the two methods. In most cases they are minor, with a high correlation ($R^2 > 0.93$) and no obvious bias of

the attributed changes in runoff and $E_T$. Hence, uncertainties in the input data are arguably more relevant than methodological issues.

## 6 Conclusions

A simple, yet powerful framework was used to analyze changes in the hydro-climatology of the federal state of Saxony in Germany. The large set of catchments with long term hydrological monitoring data in combination with a dense network of
meteorological stations allowed a comprehensive study on how climate and land-surface changes impacted runoff. During the last decade from 2011-20 a long and extreme drought event led to significant changes in catchment runoff, which were quantified using a decomposition method. The decomposition also highlighted the impacts of land-surface changes in a number of catchments revealing two main and spatially distinct patterns. First, recovery from past forest disturbance leads to a reduced runoff and intensifies the already existing negative trend due to a lack in precipitation. Second, impacts of the drought on
vegetation health are severe and effect the water balance decisively. This is especially relevant for forested regions which have experienced huge amounts of damaged wood due to bark beetle infestations reducing catchment $E_T$.

We conclude that the hydro-climatology of Saxony has become rather non-stationary leading to methodological and practical challenges for water resources management. Saxony is located in a region where climate models predict no clear trend. It is in the transition of the "dry getting drier and the wet getting wetter" paradigm. The past years point out, that this does not mean
there are no changes. Instead we may prepare for more variation with extended phases of dry and hot conditions and increasing number of intense rainfall events, which will affect water redistribution and thereby the average fresh water availability.

*Code and data availability.* Aggregated data and code to perform analysis and diagrams can be acquired from the author.

*Author contributions.* MR conceived the study, acquired data and performed analyses. MR and CH discussed the results and drafted the outline of the manuscript. CH provided information on anthropogenic influences on catchment water budgets. MR wrote the first draft which
was reviewed and edited by CH.

*Competing interests.* The authors have no competing interests.

*Acknowledgements.* We thank all data providers for sharing their data. In particular the REKIS.org initiative which collects meteorological data from different services for Saxony and bordering countries. MR is grateful to Rico Kronenberg (TU Dresden) for providing the daily data and Philipp Körner (IAMK GmbH, Dresden) for discussion on reference evapotranspiration. For the hydrological data we thank the
Landesamt für Umwelt, Landwirtschaft und Geologie (LfULG) for providing most of their discharge data on the website as well as Heike Mitzschke (Landeshochwasserzentrum, LfULG) for sending additional data and explanations on river gauge location data. Further we thank Ralf Haupt (Hochwassernachrichtenzentrale, Thüringer Landesamt für Umwelt, Bergbau und Naturschutz) for providing the data of gauge





Greiz/Weiße Elster. Finally MR thanks Christian Bernhofer and Thomas Plunkte (TU Dresden) for discussions on ongoing changes related to the Wernersbach research catchment, which raised the motivation to extend the previous study from 2014.





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

**Figures**

**Supplementary materials**

**Catchment data**

| | station / river | major basin | gaugeID | elev | area | forest | damage | $P$ | $E_0$ | $R$ | $E_T$ | miss |
|---|---|---|---|---|---|---|---|---|---|---|---|---|
| 2 | Buschmuehle/Kirnitzsch | Upper Elbe | 550090 | 397 | 98 | 78 | 1 | 854 | 683 | 296 | 565 | 20 |
| 3 | Kirnitzschtal/Kirnitzsch | Upper Elbe | 550110 | 379 | 154 | 77 | 0 | 848 | 688 | 289 | 559 | 0 |
| 4 | Porschdorf/Lachsbach | Upper Elbe | 550190 | 378 | 267 | 39 | 0 | 850 | 691 | 362 | 487 | 0 |
| 5 | Sebnitz/Sebnitz | Upper Elbe | 550302 | 422 | 102 | 47 | 1 | 877 | 681 | 438 | 450 | 19 |





| | | | | | | | | | | | | |
|---|---|---|---|---|---|---|---|---|---|---|---|---|
| 6 | Neustadt/Polenz | Upper Elbe | 550390 | 394 | 40 | 32 | 0 | 859 | 687 | 417 | 451 | 21 |
| 7 | Markersbach/Bahra | Upper Elbe | 550710 | 545 | 49 | 45 | 10 | 805 | 657 | 364 | 439 | 20 |
| 8 | Bischofswerda/Wesenitz | Upper Elbe | 550801 | 363 | 69 | 30 | 0 | 823 | 692 | 354 | 470 | 19 |
| 9 | Elbersdorf/Wesenitz | Upper Elbe | 550810 | 315 | 227 | 17 | 0 | 793 | 703 | 297 | 496 | 0 |
| 10 | Dohna/Müglitz | Upper Elbe | 550940 | 557 | 198 | 35 | 6 | 840 | 652 | 403 | 437 | 1 |
| 11 | Geising/Rotes Wasser | Upper Elbe | 550961 | 780 | 26 | 53 | 35 | 957 | 602 | 537 | 426 | 16 |
| 12 | Kreischa/Lockwitzbach | Upper Elbe | 551000 | 380 | 44 | 23 | 0 | 766 | 692 | 252 | 517 | 14 |
| 13 | Klotzsche/Prießnitz | Upper Elbe | 551100 | 261 | 40 | 60 | 0 | 732 | 720 | 267 | 472 | 20 |
| 14 | Rehefeld/Wilde Weißeritz | Upper Elbe | 551302 | 808 | 15 | 91 | 61 | 974 | 597 | 754 | 218 | 12 |
| 15 | Ammelsdorf/Wilde Weißeritz | Upper Elbe | 551310 | 734 | 49 | 64 | 20 | 966 | 614 | 599 | 374 | 2 |
| 16 | Beerwalde/Wilde Weißeritz | Upper Elbe | 551320 | 663 | 81 | 53 | 12 | 940 | 630 | 519 | 422 | 6 |
| 17 | Baerenfels/Pöbelbach | Upper Elbe | 551510 | 742 | 6 | 75 | 12 | 983 | 611 | 746 | 261 | 20 |
| 18 | Freital/Poisenbach | Upper Elbe | 551561 | 295 | 12 | 42 | 0 | 701 | 715 | 216 | 490 | 26 |
| 19 | Piskowitz/Ketzerbach | Upper Elbe | 552011 | 214 | 157 | 0 | 0 | 660 | 727 | 122 | 542 | 25 |
| 20 | Seerhausen/Jahna | Upper Elbe | 552110 | 178 | 153 | 0 | 0 | 627 | 734 | 58 | 571 | 16 |
| 21 | Merzdorf/Döllnitz | Upper Elbe | 552210 | 167 | 211 | 7 | 0 | 611 | 738 | 135 | 475 | 0 |
| 22 | Zescha/Hoyersw. Schwarzwasser | S. Elster | 554220 | 226 | 180 | 13 | 0 | 704 | 721 | 181 | 527 | 15 |
| 23 | Pietzschwitz/Langes Wasser | S. Elster | 554260 | 255 | 42 | 12 | 0 | 726 | 716 | 232 | 498 | 12 |
| 24 | Koenigsbrueck/Pulsnitz | S. Elster | 554420 | 278 | 92 | 30 | 0 | 755 | 713 | 272 | 484 | 10 |
| 25 | Grossdittmannsdorf/Große Röder | S. Elster | 554520 | 247 | 300 | 31 | 0 | 739 | 721 | 238 | 500 | 0 |
| 26 | Kleinraschuetz/Große Röder | S. Elster | 554550 | 198 | 679 | 25 | 0 | 683 | 731 | 187 | 494 | 11 |
| 27 | Golzern/Vereinigte Mulde | Mulde | 560021 | 481 | 5442 | 31 | 4 | 840 | 671 | 354 | 486 | 0 |
| 28 | Bad Düben/Vereinigte Mulde | Mulde | 560051 | 439 | 6171 | 30 | 3 | 811 | 679 | 319 | 490 | 10 |
| 29 | Zwickau/Zwickauer Mulde | Mulde | 562070 | 633 | 1030 | 56 | 2 | 953 | 637 | 434 | 519 | 0 |
| 30 | Wechselburg/Zwickauer Mulde | Mulde | 562115 | 491 | 2107 | 35 | 1 | 851 | 669 | 392 | 459 | 0 |
| 31 | Muldenberg/Rote Mulde | Mulde | 563000 | 769 | 5 | 99 | 0 | 986 | 609 | 628 | 337 | 26 |
| 32 | Sachsengrund/Große Pyra | Mulde | 563290 | 927 | 7 | 100 | 12 | 1151 | 571 | 798 | 340 | 22 |
| 33 | Aue/Schwarzwasser | Mulde | 563790 | 744 | 362 | 68 | 6 | 1014 | 615 | 548 | 466 | 0 |
| 34 | Niedermuelsen/Mülsenbach | Mulde | 564201 | 364 | 50 | 20 | 0 | 753 | 693 | 276 | 482 | 22 |
| 35 | Niederlungwitz/Lungwitzbach | Mulde | 564300 | 353 | 138 | 11 | 0 | 744 | 700 | 310 | 445 | 18 |
| 36 | Harthau/Würschnitz | Mulde | 564620 | 435 | 136 | 17 | 0 | 819 | 688 | 336 | 491 | 19 |
| 37 | Berthelsdorf/Freiberger Mulde | Mulde | 566010 | 597 | 243 | 27 | 1 | 928 | 646 | 444 | 484 | 0 |
| 38 | Nossen/Freiberger Mulde | Mulde | 566040 | 486 | 585 | 20 | 0 | 858 | 670 | 368 | 489 | 0 |
| 39 | Erlln/Freiberger Mulde | Mulde | 566100 | 504 | 2983 | 30 | 6 | 851 | 666 | 376 | 482 | 19 |
| 40 | Wolfsgrund/Chemnitzbach | Mulde | 567000 | 627 | 37 | 13 | 0 | 946 | 639 | 556 | 389 | 0 |
| 41 | Tannenberg/Zschopau | Mulde | 567400 | 663 | 91 | 38 | 0 | 960 | 632 | 509 | 448 | 9 |
| 42 | Hopfgarten/Zschopau | Mulde | 567420 | 703 | 529 | 46 | 11 | 933 | 624 | 468 | 465 | 0 |
| 43 | Lichtenwalde/Zschopau | Mulde | 567451 | 619 | 1575 | 44 | 11 | 907 | 642 | 431 | 476 | 0 |
| 44 | Annaberg/Sehma | Mulde | 567590 | 753 | 49 | 47 | 2 | 959 | 615 | 499 | 458 | 17 |
| 45 | Pockau/Flöha | Mulde | 568140 | 689 | 385 | 63 | 27 | 920 | 625 | 469 | 454 | 8 |
| 46 | Borstendorf/Flöha | Mulde | 568160 | 662 | 644 | 57 | 19 | 918 | 631 | 438 | 480 | 0 |
| 47 | Deutschgeorgenthal/Rauschenbach | Mulde | 568200 | 743 | 10 | 84 | 14 | 983 | 616 | 549 | 443 | 16 |
| 48 | Neuwernsdorf/Wernsbach | Mulde | 568250 | 757 | 7 | 94 | 68 | 983 | 610 | 589 | 401 | 17 |
| 49 | Rauschenbach/Rauschenfluß | Mulde | 568300 | 741 | 7 | 96 | 48 | 958 | 622 | 642 | 328 | 15 |
| 50 | Rothenthal/Natzschung | Mulde | 568350 | 768 | 75 | 81 | 47 | 911 | 610 | 566 | 344 | 0 |
| 51 | Zoeblitz/Schwarze Pockau | Mulde | 568400 | 706 | 129 | 58 | 15 | 924 | 623 | 531 | 390 | 1 |
| 1 | Klingenthal/Zwota (Svatava) | W. Elster | 530020 | 721 | 59 | 78 | 2 | 990 | 616 | 605 | 386 | 11 |
| 52 | Adorf/Weiße Elster | W. Elster | 576400 | 599 | 171 | 57 | 0 | 793 | 643 | 303 | 485 | 20 |
| 53 | Oelsnitz/Weiße Elster | W. Elster | 576410 | 574 | 328 | 50 | 0 | 792 | 649 | 297 | 496 | 10 |
| 54 | Magwitz/Weiße Elster | W. Elster | 576420 | 558 | 376 | 44 | 0 | 780 | 652 | 281 | 499 | 0 |
| 55 | Strassberg/Weiße Elster | W. Elster | 576421 | 532 | 611 | 37 | 0 | 742 | 657 | 258 | 491 | 15 |
| 56 | Greiz/Weiße Elster | W. Elster | 576470 | 474 | 1255 | 31 | 0 | 732 | 670 | 260 | 471 | 0 |





| 57 | Hasenmuehle/Trieb | W. Elster | 577100 | 532 | 100 | 39 | 0 | 816 | 659 | 297 | 525 | 17 |
| 58 | Mylau/Göltzsch | W. Elster | 577220 | 515 | 155 | 34 | 0 | 839 | 661 | 374 | 465 | 0 |
| 59 | Streitwald/Wyhra | W. Elster | 577901 | 242 | 178 | 8 | 0 | 662 | 717 | 153 | 510 | 4 |
| 60 | Leipzig/Parthe | W. Elster | 578110 | 144 | 315 | 14 | 0 | 606 | 735 | 93 | 512 | 0 |
| 61 | Schirgiswalde/Spree | Spree | 582010 | 382 | 179 | 34 | 2 | 806 | 688 | 350 | 460 | 12 |
| 62 | Wuischke/Wuischker Wasser | Spree | 583140 | 403 | 3 | 91 | 0 | 759 | 689 | 305 | 447 | 28 |
| 63 | Kotitz/Kotitzer Wasser | Spree | 583170 | 252 | 29 | 4 | 0 | 686 | 714 | 191 | 490 | 22 |
| 64 | Oehlisch/Schwarzer Schöps | Spree | 583230 | 275 | 35 | 10 | 0 | 681 | 714 | 186 | 495 | 27 |
| 65 | Holtendorf/Weißer Schöps | Spree | 583280 | 272 | 54 | 12 | 0 | 685 | 715 | 189 | 498 | 5 |
| 66 | Saerichen/Weißer Schöps | Spree | 583290 | 243 | 135 | 9 | 0 | 678 | 722 | 187 | 494 | 12 |
| 67 | Hartau/Lausitzer Neiße | Neiße | 660010 | 483 | 376 | 45 | 4 | 896 | 663 | 488 | 419 | 7 |
| 68 | Zittau/Lausitzer Neiße | Neiße | 660100 | 443 | 686 | 36 | 3 | 844 | 671 | 406 | 445 | 5 |
| 69 | Seifhennersdorf/Mandau | Neiße | 662001 | 419 | 75 | 22 | 4 | 815 | 676 | 378 | 445 | 17 |
| 70 | Rennersdorf/Pließnitz | Neiße | 663000 | 331 | 79 | 25 | 0 | 709 | 698 | 211 | 494 | 20 |
| 71 | Rennersdorf1/Petersbach | Neiße | 663100 | 328 | 64 | 25 | 0 | 707 | 698 | 219 | 483 | 20 |

Table 1: River stations analyzed over the period 1951 - 2020. The column $elev$ denotes the mean catchment elevation in meters above sea level, $area$ denotes catchment area in $km^2$, $forest$ gives the relative coverage of forest land use (%) based on Corine 1990 data. Forest damage in percent is given in the "damage" column, using the Corine class 324. The columns $P$, $E_0$, $R$ and $E_T$ denote average annual water balance components for the basins in mm/yr. $miss$ gives the number of missing years.

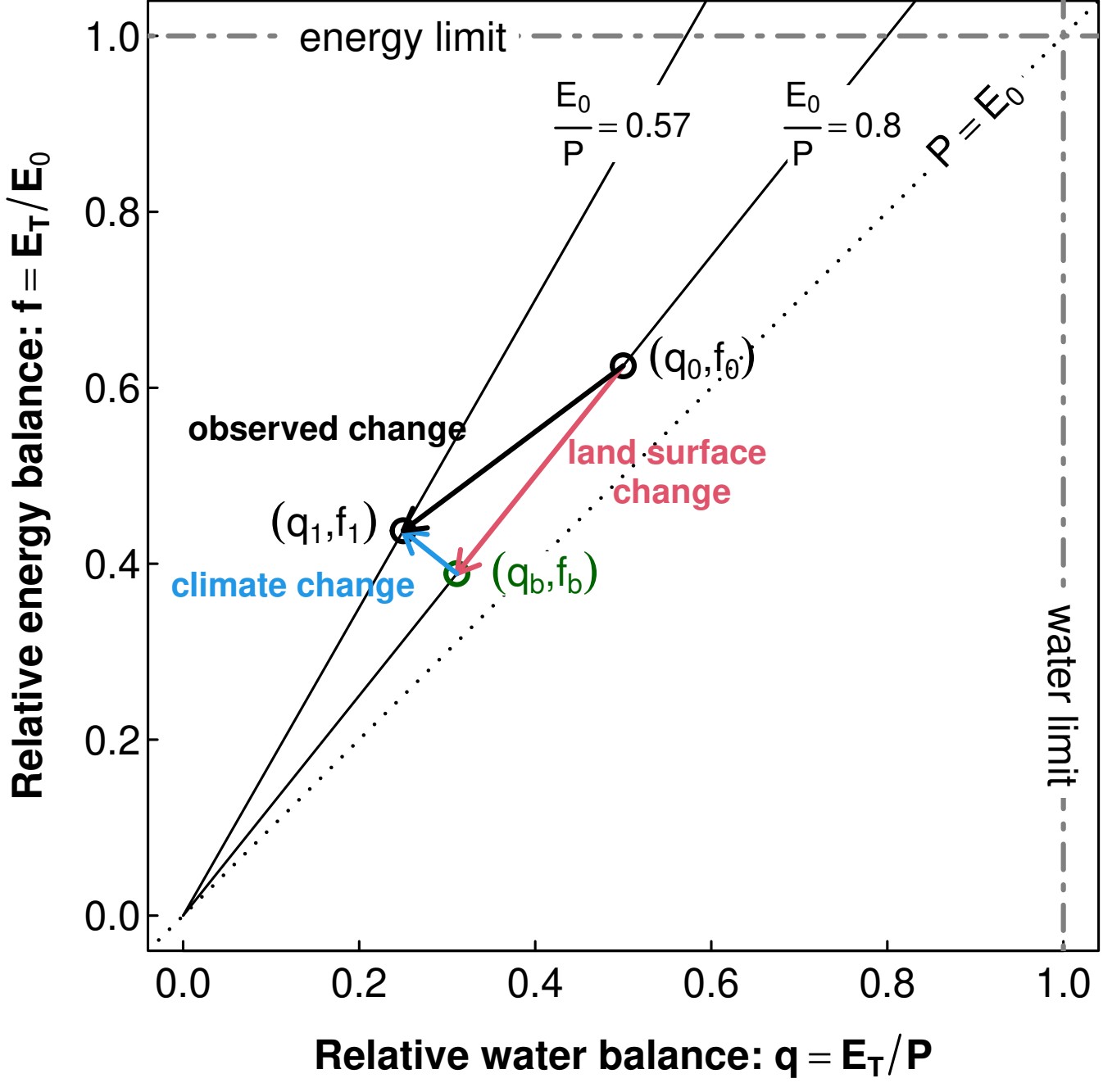

**Figure 1.** Illustration of the separation of climate and land surface changes using a relative energy to relative water balance diagram. The example shows two hydroclimatic states before $(q_0, f_0)$ and after transition $(q_1, f_1)$. The position of point $(q_b, f_b)$ is determined by using the described geometric approach (3). The bold arrows depict the climatic and the land surface components of this transition. For illustration we used case conditions as a reference: $P_0 = 1000$ mm/yr, $E_{0,0} = 800$ mm/yr, $E_{T,0} = 500$ mm/yr and a state after hypothetical climatic and land-surface change with $P_1 = 1400$ mm/yr, $E_{0,1} = 800$ mm/yr, $E_{T,1} = 350$ mm/yr. Thereby $E_T$ decreased by 30 %.

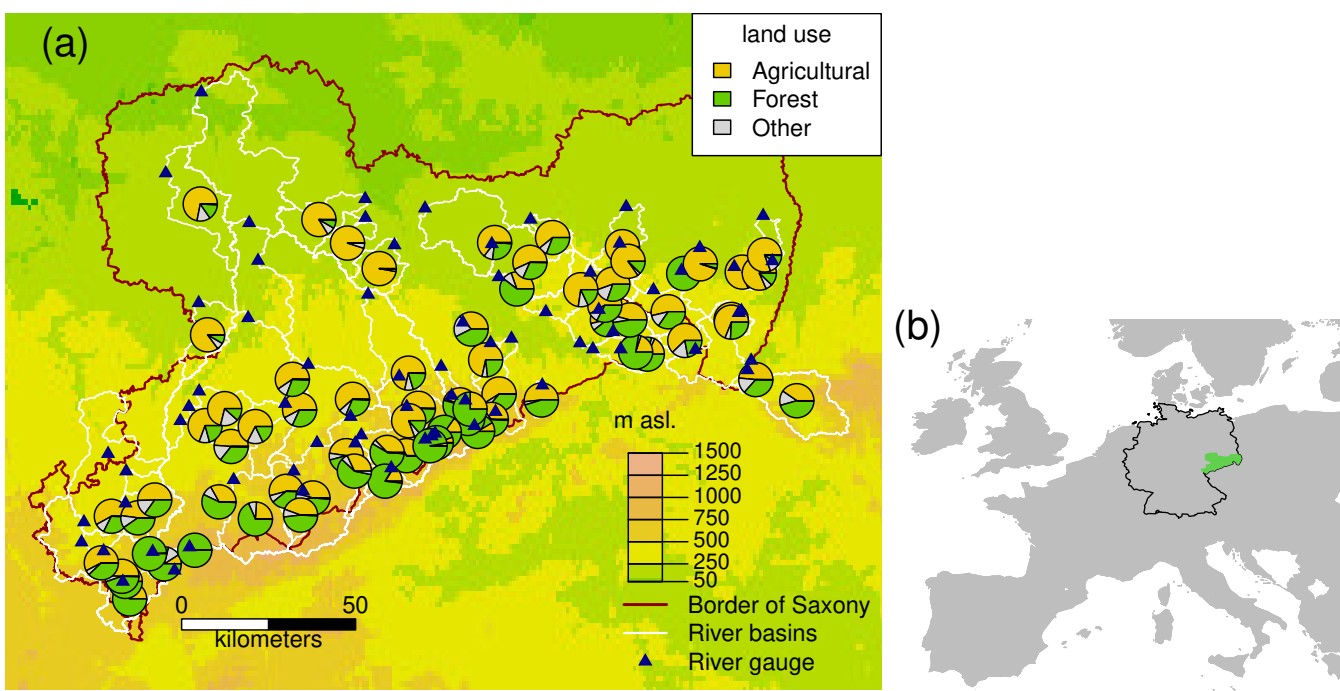

**Figure 2.** Map of the federal State of Saxony, its topography, catchment boundaries and river discharge measurement stations analysed in this study. Land-use fraction depicted as pie chart of each catchment is derived from CORINE Land-cover data from the year 2000. Panel b) shows the location of Saxony with Germany and Europe for orientation using polygon data from NUTS 2021 (ec.europa.eu).



**Figure 3.** Maps of long term (1951–2020) annual mean **(a)** precipitation, **(b)** annual potential evapotranspiration (FAO-56 grass-reference evapotranspiration, (Allen et al., 1994)), **(c)** annual runoff and **(d)** the residual water budgets ($E_T = P - R$). Observation stations which have been used to derive spatial fields of $P$ and $E_0$ are depicted as circles with diameter corresponding to the number of years of available data. The maps of mean annual precipitation $P$ and the FAO reference potential evapotranspiration $E_0$ have been derived by averaging the individual annual raster maps used to calculate the basin averages. Note, that panels (a) and (c) as well as (b) and (d) use the same color scale respectively.





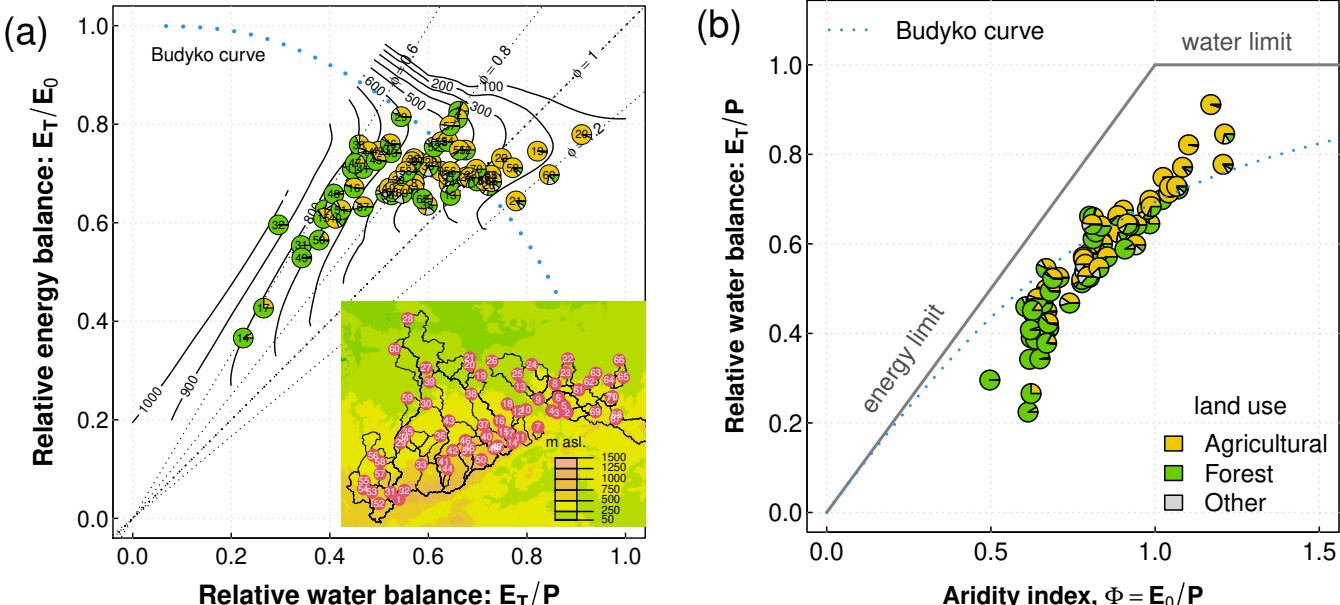

**Figure 4.** Long term catchment water and energy partitioning. The pie-charts show the areal percentage of land use of each basin. In the right panel, the average basin elevation is used to predict the contour lines using local polynomial (LOESS) regression. This demonstrates the general height dependency of the basins climate. Further, the transition from wet basins with high runoff ratio to lower values is also reflected by land-use. The inset shows a map of the catchments, its gauge id number and altitude as background raster.

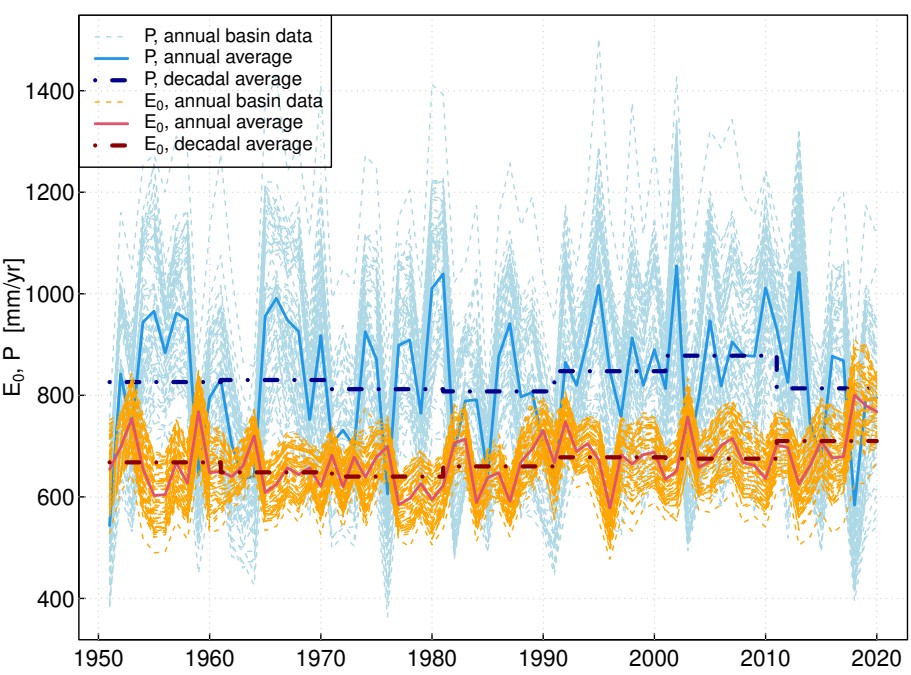

**Figure 5.** Time series of annual $P$ and $E_0$ for all basins (thin dashed lines), cross basin annual average time series (full lines) and the all basin decadal averages (dot-dashed lines).



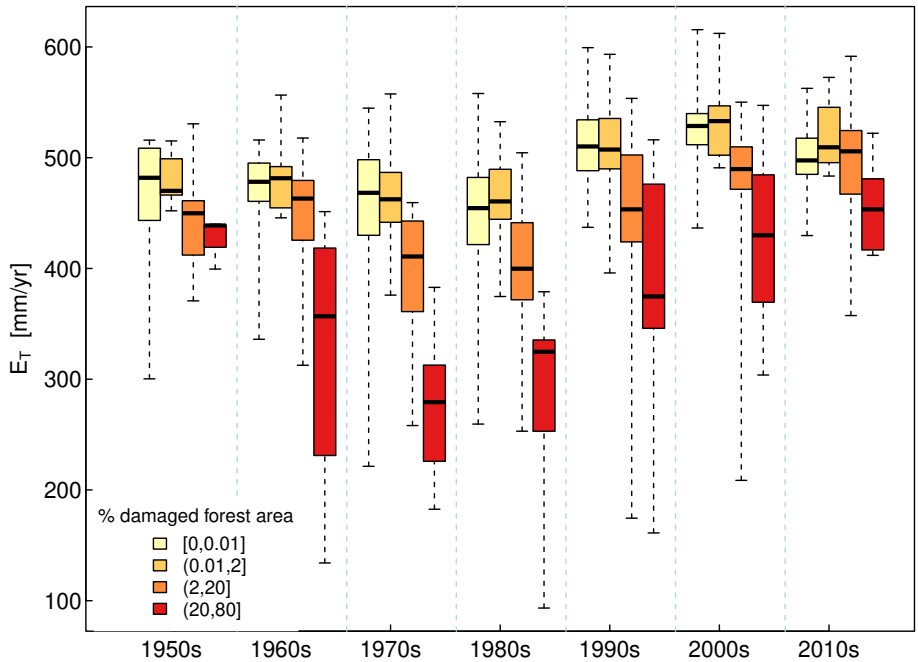

**Figure 6.** Boxplots of decadal averages of $E_T$ for four groups of catchments with different ranges of damaged forest area per catchment area using the Corine 1990 class 324 (transitional scrub forest).

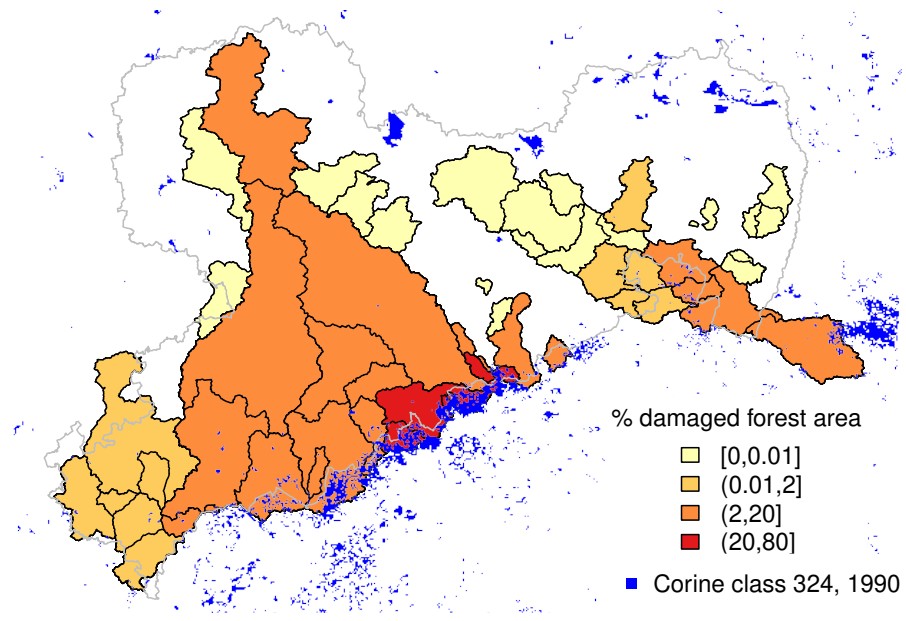

**Figure 7.** Map of forest damage using Corine 1990 class 324 (transitional scrub forest). Catchments are grouped by the relative area of forest damage.



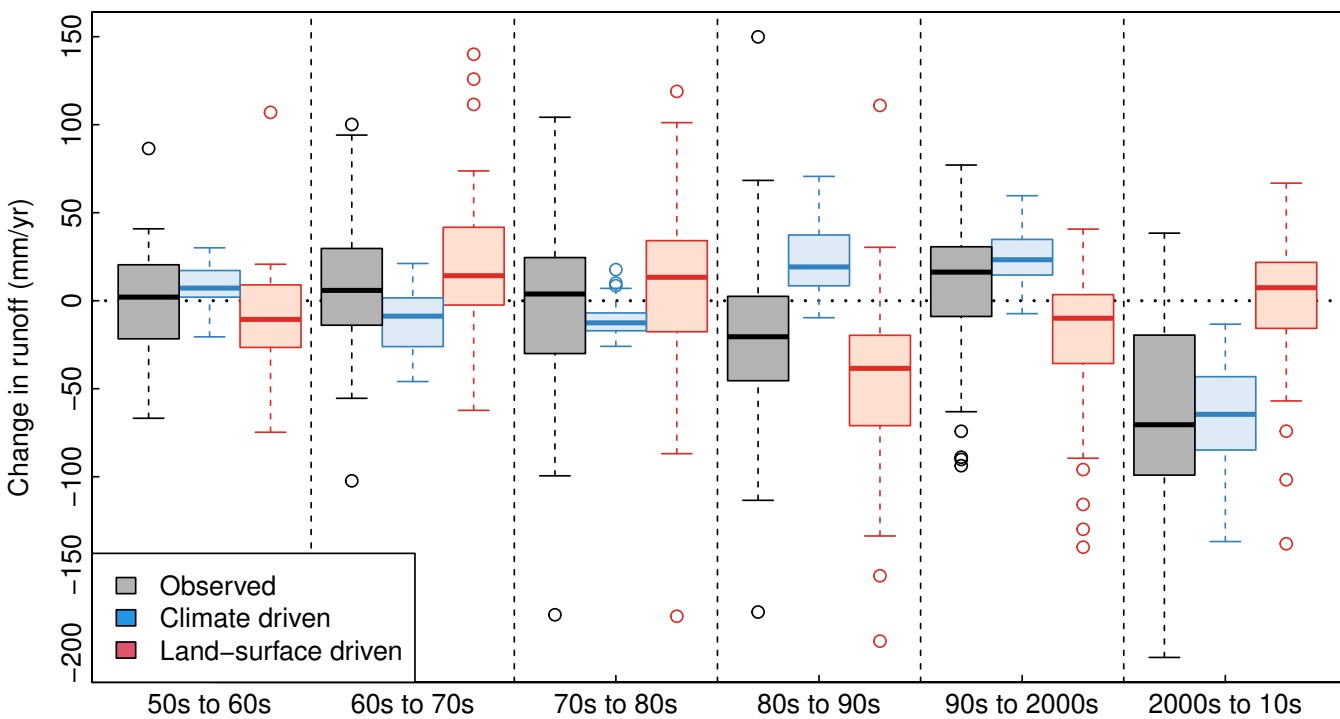

**Figure 8.** Decadal changes in annual mean catchment runoff across all catchments. Box-whisker plots are shown for observed (gray), climate (blue) and land-surface attributed runoff changes using the decomposition method introduced in the methods section. The boxes show the inter-quartile range (IQR) with the bold horizontal line being the median of the distribution. The whiskers show the outer ranges, with points marking outliers outside a 1.5 IQR range.





**Figure 9.** Annual averages per decade of the water and energy partitioning ratios for each basin analyzed. The arrows denote the change from one decade into the following decade. There is one panel for each decadal change, starting in 1950s in panel (a) to 2000s to 2010s in panel (f). Significant changes are marked with bold arrows and a red border of the respective catchment in the map inset. The inlay maps show the location of the river gauges, the spatial extent of forest damages and since 1990 blue pixels show the Corine transitional forest class.



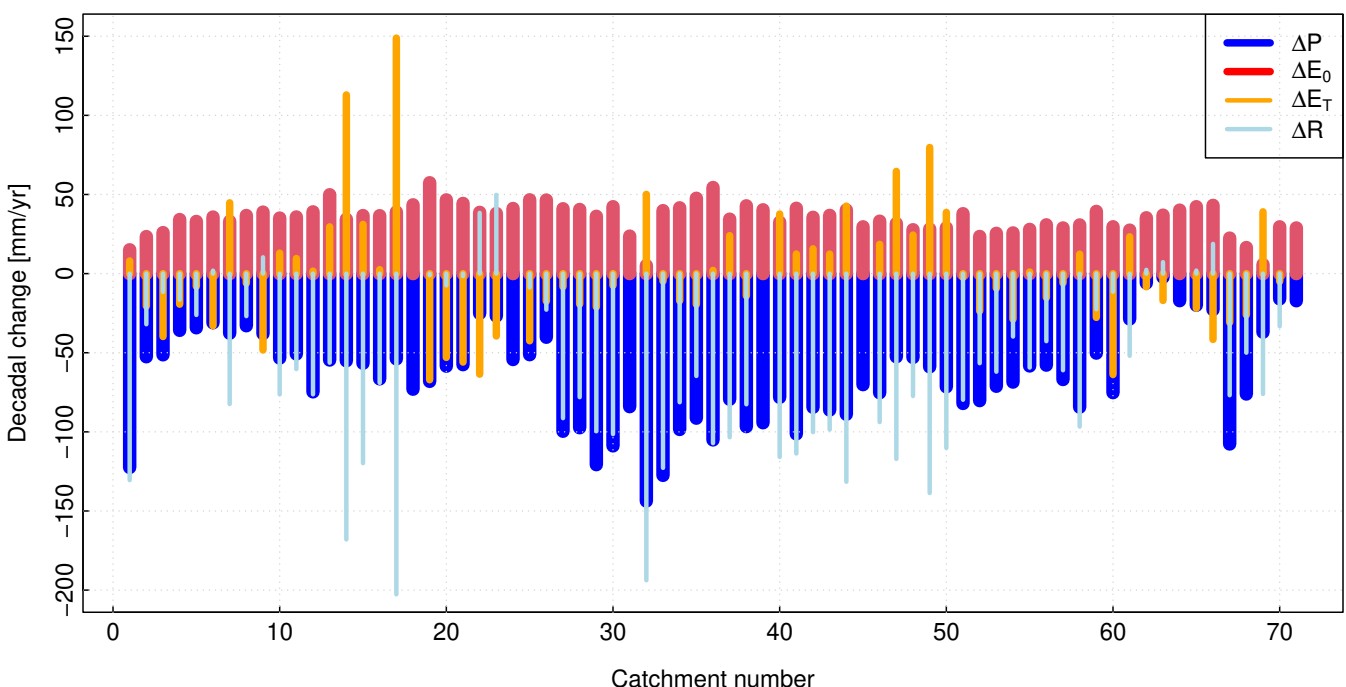

**Figure 10.** Change in decadal water balance components from the decade 2001-10 to 2011-20 in mm/yr for each catchment.

**Figure 11.** Maps of decadal changes (2001-10 to 2011-20) in **(a)** precipitation and runoff, **(b)** potential evaporation (FAO-56 grass-reference evapotranspiration, (Allen et al., 1994)) and catchment evapotranspiration ($E_T = P - R$), **(c)** climate attributed changes in runoff, and **(d)** land-surface attributed changes in runoff. While panels **(a)** and **(b)** visualize the changes on the catchment level and at the gauging stations (triangles), panels **(c)** and **(c)** use only the catchment area to present changes.





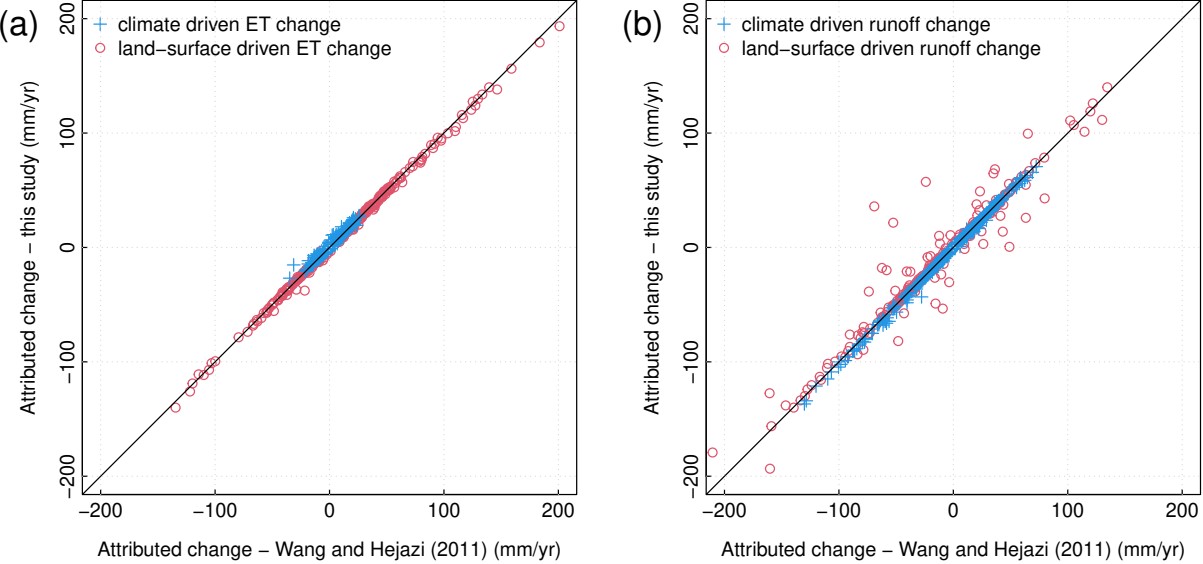

**Figure 12.** Comparison of the decomposition method of this study with the method of Wang and Hejazi (2011) using the Mezentsev (1955) parametric Budyko function to separate climate and land-surface induced changes on $E_T$ (panel a) and runoff (b). Here all decadal transitions of all catchments with sufficient data are used for the comparison.