# Peer review of "Impacts of climate and land-surface change on catchment evapotranspiration and runoff from 1951-2020 in Saxony, Germany"

_Hydrology and Earth System Sciences, 2024_

## Referee Comment (RC3)

To whom it may concern,

Thank you for allowing me the opportunity to review your paper. From my understanding, your paper is trying to tell the following story:

- Annual runoff data is changing due to climate change, and Saxony, Germany, has records that need to be upgraded.
- This paper updates the hydro-climatology for 71 catchments in Saxony and extends the previous study period to cover 1951-2020.
- The paper discusses water-energy partitioning over multiple decades.
- The method used is compared to others proposed by the literature.

Overall, the paper is clear and concise and merits publication. I propose a publication with minor technical corrections (see below).

Best,
Tadd Bindas

Minor Comment:
- Page 30 has a formatting issue where the figure is too large to fit on a page. This is a technical correction

---

## Author Response (AR1)

**1 Authors response**

Overview of all comments from referees, our response and changes in the manuscript. Responses are formatted in bold. Changes in italic. This overview uses the individual replies from the interactive discussion.

**1.1 Author reply to Reviewer 1**

Thiago Victor Medeiros do Nascimento, 04 Feb 2024.

*General comments*

The manuscript presents an interesting and relevant work addressing scientific questions that are within the scope of HESS. The main objective of the authors is to analyse and separate the effects of climate and landscape change in several catchments over Saxony, Germany. In the paper the authors present what is at the same time one novelty and an update of a prior analysis performed by Renner et al. (2014). The manuscript is well-structured, clear, and concise. The introduction is well written, the methodology is clearly stated, and the results are well presented. The figures present a high quality-level with only minor needed-improvements. Taking into consideration that the present manuscript presents enough novelty and a rich discussion, I suggest that it deserves to be published after some minor corrections and adaptations regarding the quality of the discussion and figures.

**Author reply: thank you for the encouraging summary of our work and hints to improve which we will consider during revision.**

*Specific comments* In this section I present some specific comments to be answered and implemented in the manuscript. The authors assume of disregarding GW recharge and point P-R as a valid assumption for ET. Could you please discuss it better in the text the reasoning why this is a valid assumption to your study area?

**Author reply: We estimate catchment evapotranspiration by using the simple water balance equation $P = E_T + R + \Delta S_w$. In order to apply $E_T = P - R$, we need to reduce the effect of storage changes such as soil water dynamics and also groundwater recharge. To make this clear we added two sentences in the methods section 2.1:** *When integrating over time the change in water storage in catchments $\Delta S_w$ should converge to zero (Dyck and Peschke, 1995) and one can estimate catchment evapotranspiration by $E_T = P - R$. However, 5-10 years of data are recommended to reduce water storage effects (Zhang et al., 2001).*

**For the majority of catchments in this study this approach is suitable since most catchments are situated in regions with low conductive bedrock, where groundwater storage and variations in groundwater recharge are of minor importance. However, in northern Saxony which is more flat there are larger groundwater aquifers and catchments in these areas actually show significant cross correlation of annual runoff and precipitation at a lag of one year, see (Renner et al.,**

2014). In such regions there could still be an imbalance between groundwater recharge and release even at the decadal time scale considered in this study. This limitation is reported in the discussion section of the manuscript. It also highlights an important research question for follow-up studies.

What are the criteria used to use the thresholds of 2% and 20% to define minor and major forests damages? (L247-249)

**Author Reply: We used the same thresholds as used in the Renner et al., 2014 paper. These have been selected by the following criteria. Major forest damages > 20% of catchment area is a criteria taken from the literature. The paired catchment review paper of Bosch and Hewlett (1980) reports significant streamflow increases after clear-cut when more than 20% where affected. The threshold category of 2%-20% was applied to select catchments where some measurable area was affected. The exact value was also set to have sufficient number of catchments per group, which is rather subjective, however, note the spatial coherence as seen in the map in Figure 7.**

We added the following text in the data section: *Based on this data Renner et al. (2014) calculated the relative affected area for each catchment and classified the catchments in groups with different magnitudes of forest damage. The class major forest damages with a threshold of > 20% of catchment area was selected following Bosch and Hewlett (1982)a who reported significant streamflow increases in paired catchment studies after clear-cuts when more than 20% where affected. The threshold category of 2%-20% was applied to select catchments where a measurable area was affected. Renner et al. (2014) found that 38% of the catchments have no forest damage ...*

*Technical corrections* In this section I present some technical corrections and suggestions to be implemented in the writing and figures of the text. This section is divided by each text section (i.e., abstract, introduction)

*Figures* Figure 2: Please add the north arrow in the main map.

**Author Response: Arrow was added.**

Figure 3: if you intend to present this figure as a map, please consider adding their north arrow and scale bar.

**Author Response: Arrow and scale bar were added to panel c).**

Figure 4: use the letters "a", "b" and "c" in the figure title to improve the description of the subplots. Finally, in the figure title, when you mention "in the right panel" did you mean "in the right panel within subplot a? Using the letters as already suggested will dimmish these misunderstandings.

**We updated the figure caption to improve the description of the subplots.**

Figure 7: The map lacks grid, scale bar and north. **Author Response: Grid, Arrow and scale bar were added**

Figure 10: As you classify in L312-323 the catchments in three groups (i.e., I, ii and iii), why do not you group these catchments in Figure 10? This would improve the visualization and discussion of your results.

**Author Reply: Thank you for the good suggestion. We added letters for the three categories in Figure 10. Note that we changed the order of the figures following reviewer 2.**

Figure 11: The figures lack north and scale bar.

**Author Response: Arrow and scalebar were added to panel c).**

*1 introduction* The introduction is well structured. However, as a suggestion I would consider the addition of at least two/three recent studies describing the effects of environmental changes (climate and landscape changes) in other study areas. This is one suggestion to enrich your already well-organized text.

**We feel that this is covered in the introduction where we have a whole paragraph on runoff trends and citing research which attributed these to various environmental changes, see L20-29.**

L54-L59: I understand that this is your objective paragraph. I would consider making it clearer that the present study also works updating and extending the previous study of Renner et al. (2014). Please rephrase this first paragraph sentence to make it clear to the reader. Something like: "This study extends and updates the work of Renner et al. (2014) (...)".

**Thank you. The first sentence of the paragraph was updated.**

2 Methods 2.1. Catchment water and energy balance

L68-69: Could you please enrich the discussion about why is this a fair assumption considering your study? The addition of one reference would be enough to enrich and self-guard your methodology against potential critics.

**We added the reference to (Fan, 2019) which discusses this problem.**

3 Data 3.1. Hydrological data

L153: Could you add the link of the website?

**Done**

L158: Why did you use a threshold of 20 non-missing days in a month? Could you make it clear why did you chose to work with this threshold? **We have set this threshold to ensure that at least 67% of discharge data per month were available to compute an monthly average. Using only months with complete data could reduce the length of the observation period drastically.**

L165: Due to the current size, it would be necessary to move Table 1 to supplementary materials. However, the authors could consider presenting a subset of the table in the manuscript.

**Author reply: The table is now part of the appendix**

4 Results

4.1 Long-term hydro-climatology of Saxony

L205: Please also refer to Figure 3 once you correlate elevation here. Rephrase to: "(...) (see Fig. 2 and Fig. 3b)."

**Done**

L210: I believe that this assumption needs to be better discussed in the methodology/discussion sections.

**True, we added a paragraph in the methods section.**

L221: "four general examples" instead of "three general examples", right? Since you have 0.6, 0.8, 1 and 2.

**True, is corrected.**

4.2 Annual and decadal dynamics in joint water and energy balance 4.2.1 Variability of catchment evapotranspiration

Please consider switch the position of Fig. 6 and Fig. 7. Probably showing the map before the boxplots would be more adequate for this analysis. Please consider all the needed alterations in the text if this switch is implemented,

**We moved the map and its description into section Land-use and forest damage data. The corresponding alterations in the text have been made.**

4.4 Decadal dynamics of water-energy partitioning from 1951 to 2000

L288: please refer the figure here (e.g., Fig. 9X).

**Done**

4.5 Decadal dynamics of water-energy partitioning from 2000 to the 2010s

L345: Rephrase to: "the already discussed decomposition (. . . ) ".

**Done.**

5 Discussion 5.1 Limitations 5.1.2 Catchment integration and groundwater transfers

Please consider the already mentioned comment regarding the improvement of the discussion regarding the groundwater simplification in the "specific comments" section of this document.

6 Conclusions

L473: Please do not use the term "A simple (. . . )" to refer to your current work.

**Done**

**1.2 Author reply to Reviewer 2**

Anonymous Referee #2, 05 Feb 2024, https://doi.org/10.5194/hess-2024-6-RC2

The manuscript by Renner & Hauffe analyzed the temporal dynamics from 1951 to 2020 of water and energy balance for catchments in Saxony, Germany. They analyzed the changes in evapotranspiration and runoff, based on partitioning between climate and land-use induced changes. The study showed different responses of runoff for distinct catchments, and attributed it to different land use changes (e.g., forest regrowth vs vegetation damages). Overall, I think the manuscript is good. The paper is well-written, well-structured, and the conclusions are supported by the analyses. However, the study is methodologically very similar to the one by Renner et al. (2014). The main difference is that Renner et al. (2014) is limited until 2009, and the current manuscript goes up until 2020. This raises three comments: 1 – I believe that it needs to be more clearly stated that this study is a "continuation" of Renner et al. (2014) (e.g., earlier in the Introduction section).

**Author reply: also reviewer 1 recommended this, so we added a sentence in the introduction.**

2 – The innovative aspect of including the recent decade could be highlighted (e.g., what is the added value of analyzing the more recent decade? What has changed?). If nothing has changed much, I think it would have been a poor contribution just "continuing" the same study for the longer time series. From what I understood in Fig 9, it looks like in the past decades, land-use change was the main influence for runoff/ET changes (i.e., arrows roughly parallel to diagonal line), whereas in the recent decade, climate change started to play a more important role in impacting runoff/ET (i.e., arrows roughly orthogonal to diagonal line). Please correct me if I understood this wrong. I think that this is one of the main contributions from the manuscript, and comments related to it could be added to Introduction/Discussion/Conclusions sections, and even the Abstract, if the authors consider it relevant.

**Author Reply: indeed the strong effect of climate change which became the strongest driver of runoff changes in the last decade is important and interesting in its own. Therefore the manuscript highlights the changes of the last 20 years in the results and discussion section. This part is also highlighted in the middle part of the abstract. A further innovative aspect is that the method was extended to attribute climate and land-cover effects also on runoff. In the earlier paper we focused only on ET.**

3 – Fig 1 and Fig 2 are VERY similar to the ones presented in Renner et al. (2014). I think maybe this needs to be clearly stated, otherwise it could indicate some degree of self-plagiarism.

**Author reply: it makes sense to re-use these figures. We now state that these figures are taken from Renner et al. (2014).**

Please also consider the following smaller concerns: In several parts of the manuscript, the authors refer to regions/rivers in Saxony (e.g., Ore Mountains, the Lausitzer Bergland, and Izer mountains in L207; Mulde river basin in L315; Elbe river, Lausitz in L341-342; L450-453). For a non-German, these regions might not be known. I would suggest adding a figure with a map and the names of these regions (at least the ones that are mentioned in the text). **Thank you for the suggestion. We added the main rivers and annotated the regions mentioned in the text.**

L11-12: "These catchments showed declines in actual evapotranspiration which could be signatures of either contributing groundwater at longer time scales" – this part about the influence of groundwater is stated in the abstract, but it is clearly mentioned in the discussions/conclusions. I would suggest either removing it from the abstract, or explaining a little bit more about it in the discussions/conclusions.

**We added a paragraph in the discussion to explain why contribution of groundwater may have sustained runoff despite the increase in aridity. Nevertheless, we deleted the sentence in the abstract.**

L21: reference in the middle of the sentence (Masseroni et al., 2021) -¿ replace it at the end of the sentence

**Done.**

L50: "Renner et al (2014) analyzed decadal variations in water and energy balance (...)" – and what were the conclusions?

**Indeed, these were missing and are now included.**

L87: "the later ratio" $->$ "the latter ratio"

**Done.**

I believe it is very hard for the reader to derive Equation (3) based on the information provided in this paper (I personally was not able to). In Renner et al. (2014), this derivation is better explained ("the angle between both vectors can be described by the scalar product divided by the vector magnitudes to give"; Eqs (3)-(5)). Either reference Renner et al (2014) for a detailed derivation, or add the details here.

**Ok, we reference to Renner et al (2014) for a detailed derivation.**

It is not clear to me either how to get to Eqs (4) and (5). Can you please provide the additional steps? (maybe this does not need to be in the final version of the paper, but I would like to be able to understand/verify it).

**Author reply: Ok, to get to equation 4 we start with the climate-induced change in evapotranspiration:**

$\Delta E_{T,C} = E_{T,1} - E_{T,b}$ **next we apply the water budget equation** $\Delta P = \Delta E_T + \Delta R$, **which yields for the climate part:** $\Delta R_C = \Delta P - \Delta E_{T,C}$, **then we can substitute** $\Delta E_{T,C}$ **to yield:** $\Delta R_C = \Delta P - E_{T,1} + E_{T,b}$. **You can rewrite** $E_{T,1} = P_1 - R_1$ **and** $\Delta P = P_1 - P_0$ **and insert both:** $\Delta R_C = P_1 - P_0 - (P_1 - R_1) + E_{T,b}$ **to yield equation 4:** $\Delta R_C = R_1 - P_0 + E_{T,b}$ .

**To get to equation 5 we start with land-cover induced change in evapotranspiration as noted in the manuscript:** $\Delta E_{T,L} = E_{T,b} - E_{T,0}$ **and also apply the water balance equation to get to the runoff changes.** $\Delta R_L = \Delta P - \Delta E_{T,L}$, **then we can substitute** $\Delta E_{T,L}$ **to yield:** $\Delta R_L = \Delta P - E_{T,b} + E_{T,0}$. **Then we rewrite** $E_{T,0}$ **and** $\Delta P$: $\Delta R_L = P_1 - P_0 - E_{T,b} + P_0 - R_0$ **and simplify to yield equation 5:** $\Delta R_L = P_1 - E_{T,b} - R_0$.

**We put the derivation in the appendix.**

L163: "runoff use?" - unclear

**The sentence was slightly re-phrased to make it clear.**

L187: "is used" -¿ "was used"

**Done.**

Fig 4: land use is provided for what year?

**The year 2000 from Corine was used in both figures.**

L239: I do not see these circles in Figure 5.

**The part of the sentence referred to a prior version. We removed it.**

Figure 5: there seems to be a white line in the legend?

**We did not see a white line in Figure 5.**

I am in general confused about the "Corine class 324" dataset. It would be good to make this clearer. First of all, are there any limitations related to using

two different datasets before and after 1990 (Section 3.3)?

**The data sets are admittedly completely different in how they estimate forest damage. Although forest authorities still do ground based measurements of forest health, their methodology changed after 1989 and are thus not easy to compare. The Corine data set derived from remote sensing covers a large area with high spatial resolution (300m) and one can derive statistics like the relative area per catchment. Therefore we decided to use the Corine data set to get the statistics for a single snapshot when the areal extent of forest damage was largest (1990). The older data set is used as qualitative illustration of the temporal evolution of forest damage compared to Corine 2000 and 2012. Since we do not perform a quantitative analysis of forest damage over time these limitations are not a problem. Given the questions by the reviewer we revised section 3.3 in the manuscript.**

In Figure 7, I do not understand the blue color (Corine class 324, 1990). I had understood that all "% damaged forest area" was based on Corine class 324, 1990.

**In Figure 7 we show the Corine class 324 raster cells in blue color which were used to estimate the relative area of forest damage per catchment. This relative area is also grouped in four categories which are shown as colored catchment boundary polygons in the map.**

In Figure 9, I find it confusing that 9d already has the Corine class 324, but it does not appear in the legend.

**Due to limited space in the subpanel the legend was missing. We fixed it in the revision.**

Also in Figure 9, the Corine dataset of the year 2000 is used (L278), but in the previous figures, it was presented for year 1990. Why the change?

**Figure 9 shows the decadal changes and to get an qualitative idea of forest damage we tried to get forest damage data of different periods (see also reply above). So in the panels of Figure 9a) to c) we used the historical ground based forest damage data sets. In 1990 we added the Corine data (Figure 9d). After 1990 we do not have the ground based data of forest damage and therefore we only show the Corine data snapshots of 2000 (Figure 9e) and 2012 (Figure 9f).**

L254-255: "From 1991–2010 all catchments have a pronounced increase in ET" – please review this statement. I am not sure I agree that catchments with low forest damage also follow this statement.

**Indeed, the statement was not correct. We now refer to those regions with moderately and heavily damaged forest.**

L327: "The widespread and large decline in annual precipitation with a median decrease of -59 mm/yr." – this sentence is a little out of context, it does not have a verb. Please re-phrase it.

**Done.**

The "limitations" section is very good. It anticipated a lot of my concerns as a reviewer. However, I would personally switch to presenting the main discussion

first, and then the limitations afterward.

**Thank you. We follow the recommendation and moved the limitations behind the main discussion.**

L477: why did it "also" highlighted?

**The word "also" has been removed.**

L480: "effect" → "affect"?

**Done.**

**1.3   Author reply to Reviewer 3**

Tadd Bindas Referee #3, https://doi.org/10.5194/hess-2024-6-RC3

Thank you for allowing me the opportunity to review your paper. From my understanding, your paper is trying to tell the following story: Annual runoff data is changing due to climate change, and Saxony, Germany, has records that need to be upgraded. **Author reply: It is not the records which need to be updated, but the water resources management has to deal with the significant changes in the observed mean annual runoff.**

This paper updates the hydro-climatology for 71 catchments in Saxony and extends the previous study period to cover 1951-2020. The paper discusses water-energy partitioning over multiple decades. The method used is compared to others proposed by the literature. Overall, the paper is clear and concise and merits publication. I propose a publication with minor technical corrections (see below). Best, Tadd Bindas

Minor Comment: Page 30 has a formatting issue where the figure is too large to fit on a page. This is a technical correction **We hope to find a good printing solution with the editorial office at HESS. In 2014 this also succeeded.**

**References**

Bosch, J. M. and Hewlett, J. D.: A review of catchment experiments to determine the effect of vegetation changes on water yield and evapotranspiration, Journal of Hydrology, 55, 3–23, https://doi.org/10.1016/0022-1694(82)90117-2, 1982.

Dyck, S. and Peschke, G.: Grundlagen der Hydrologie, 1995.

Fan, Y.: Are catchments leaky?, WIREs Water, 6, e1386, https://doi.org/10.1002/wat2.1386, 2019.

Renner, M., Brust, K., Schwärzel, K., Volk, M., and Bernhofer, C.: Separating the effects of changes in land cover and climate: a hydro-meteorological analysis of the past 60 yr in Saxony, Germany, Hydrology and Earth System Sciences, 18, 389–405, https://doi.org/10.5194/hess-18-389-2014, 2014.

Zhang, L., Dawes, W., and Walker, G.: Response of mean annual evapotranspiration to vegetation changes at catchment scale, Water Resources Research, 37, 701–708, 2001.